
# Hybrid Neural Network - Variational Data Assimilation algorithm to infer river discharges from SWOT-like data

Kevin LARNIER[(1)(2)(3)] and Jérôme MONNIER[(1)(2)]

[(1)]Institut de Mathématiques de Toulouse (IMT), France
[(2)]INSA Toulouse, France
[(3)]CS corporation, Business Unit Espace, Toulouse, France

**Correspondence:** J. Monnier (jerome.monnier@insa-toulouse.fr)

**Abstract.** A new algorithm to estimate river discharges from altimetry measurements only is designed. A first estimation is obtained by an artificial neural network trained from the altimetry large scale water surface measurements plus drainage area information. The combination of this purely data-based estimation and a dedicated algebraic flow model provides a first physically-consistent estimation. The latter is next employed as the first guess of an advanced variational data assimilation formulation. The final estimation is highly accurate for rivers presenting features within the learning partition; for rivers far outside the learning partition, the space-time variations of discharge remain accurately approximated however the global estimation presents a potential bias. Indeed, it is shown that if the estimation is based on the hydrodynamics models only, the inverse problem may be well-defined but up to a bias only (the bias scales the global estimation). This bias is removed thanks to the ANN but for rivers in the learning partition only. For rivers outside the learning partition, any mean value (eg. annual, seasonal) enables to remove the bias. Finally, the present hybrid and hierarchical inversion strategy provides more accurate estimations compared to the state-of-the-art for the considered 29 heterogeneous river portions.

## 1 Introduction

The estimation of ungauged and poorly gauged river discharges is as one of the great challenge in hydrology. Numerous satellite missions acquire every days huge amount of data of different natures (altimetry, optical etc) which may be useful to set up river flow models, see e.g. Chen and Wang (2018) and references therein. One of the ultimate goal of river flow models is to estimate the space-time varying discharge $Q(x,t)$. Setting a river flow model requires to know the bathymetry, an effective friction coefficient and the (potential) lateral fluxes, see e.g. Chow (1964). The future Surface Water and Ocean Topography (SWOT) mission (NASA-CNES et al.) planned to launch in 2022 will provide unprecedented Water Surface (WS) measurements of rivers wider than $50 - 100\ m$. SWOT instrument will measure the WS elevation $Z$ (with a decimetric accuracy over $1\ km^2$) and the WS width $W$ (with a varying uncertainty of a few $m$, depending of the river plan form). This instrument will cover a great majority of the globe with relatively frequent revisits: from 1 to 4 revisits per 21 days repeat cycle, see Rodriguez and Esteban-Fernandez (2010); Rodriguez and others (2012). Given such WS measurements and in view to set up river flow models, the following inverse problem arises: to estimate the discharge $Q(x,t)$ but also the unobserved



bathymetry $b(x)$ and a friction law parametrization $K(x,t)$. A few inversions algorithms to solve related inverse sub-problems have been developed, see e.g. Durand et al. (2016) and references therein where 5 different algorithms are compared on 19 river portions. The considered methods are based either on relatively basic flow models (the algebraic Manning-Strickler's law) or empirical hydraulic geometry power-laws. No method turned out to be accurate or robust in all considered configurations or regimes. All methods remain sensitive to the introduced priors e.g. a good knowledge of the bathymetry or the mean value

of discharge. A few data assimilation approaches based on the Kalman filter and its variants have been developed, see e.g. Biancamaria et al. (2016) and references therein. None of them consider the complete inverse problem that is infering the triplet $(Q(x,t), b(x), K(x,t))$, and not one or two of these variables only. Two approaches based on Variational Data Assimilation (VDA) (i.e. optimal control of the flow model see e.g. Asch et al. (2016) and references therein) address the complete inverse problem that is infering the full set of unknowns $(Q(x,t), b(x), K)$, see Brisset et al. (2018); Oubanas et al. (2018b, a); Larnier

et al. (2020a). In Oubanas et al. (2018a, b), the triplet of unknowns is accurately infered from the 1D Saint-Venant flow model however the priors are computed from small Gaussian perturbations of the true values of $K$ and $b(x)$. Moreover the prior defines a highly controlling rating curve $Q(Z)$ at downstream (outflow condition). As a consequence, the inversion process converges quite easily to the correct time-dependent discharge at upstream $Q_{in}(t)$ and to the corresponding bathymetry $b(x)$. Indeed, the method provides the values corresponding to the imposed rating curve which is nearly exact. The direct model

elaborated in Larnier et al. (2020a); Brisset et al. (2018) and the considered inverse problem are more advanced. Saint-Venant's dynamics flow model is considered with actually unknown downstream conditions: the normal depth are imposed and are part of the inverse problem (or $Z$ is imposed if known). Moreover this dynamics flow model is combined with an algebraic low-Froude flow model dedicated to the satellite measurements scale, see Brisset et al. (2018); Larnier et al. (2020a). This inversion strategy (implemented as the named HiVDI algorithm) enables to infer accurate space-time variations of the discharge but with

a potential bias. Applications of HiVDI algorithm have been instructive in different and complex contexts, see e.g. Tuozzolo et al. (2019); Garambois et al. (2020); Pujol et al. (2020). Comparison of this algorithm results can be found in Frasson et al. (Submitted). To be applied to worldwide ungauged rivers, no informed prior should be introduced in the inversions, neither in the direct model nor in the inverse method. This is one of the configuration investigated in Larnier et al. (2020a), however a potential bias on the obtained discharge estimation was remaining. This bias depends on the prior accuracy e.g. the mean value

of discharge or the bathymetry elevation. To our best knowledge, no investigation have been conducted to solve the present inverse problem by employing Machine Learning - Artificial Neural Network (ANN) yet. Purely-data driven estimations have been employed in hydrology, see e.g. Chen and Wang (2018) and references therein, but not to solve the present challenging inverse problem: river discharge estimations from altimetry WS measurements only.

VDA-derived solutions depend on the direct model of course but also on the prior information: covariance matrix defining

the employed metric(s) and the first guess value(s). The present covariance matrices are non-uniform therefore somehow physically-adaptive; they make improve the robustness and the accuracy of the VDA estimations compared to those in Larnier et al. (2020a). Moreover, the first guess values are here derived from a preliminary estimation of $Q(x,t)$ obtained from an Artificial Neural Network (ANN); the latter is trained from WS measurements and drainage area values. This first ANN estimation enables to next derive relatively good first values for the VDA process. These new definitions of priors (plus a few





other improvements of the VDA algorithm compared to Larnier et al. (2020a); Brisset et al. (2018)) enables to greatly improve

the estimations accuracy.

   The resulting new HiVDI (Hierarchical Variational Discharge Inference) algorithm enables to estimate very accurately the

discharge values for rivers presenting discharge values within the learning range. For rivers presenting discharges far outside

the learning range, the algorithm provides estimations with a high accuracy of space-time variations but still with a potential

bias. However the latter is much lower than those obtained in the previously mentioned studies; also this bias is removed if any

mean value (eg. seasonal, annual) is known. Moreover past a learning period of the observed rivers (typically after one year),

given newly acquired SWOT like data, the present approach provides three different estimators : the trained ANN (purely data-

driven estimator), a dynamic physically-based estimator (based on the Saint-Venant equations) enabling to extrapolate both in

space and time the estimations, and a low complexity algebraic flow model enabling real-time estimations.

This article is organized as follows. Data (altimetry and in-situ) and the three different scales of data and models are de-

tailed in Section 2. Purely data driven estimations denoted by $Q^{(ANN)}$ are analysed in Section 3, both for rivers inside the

learning partition and outside the learning partition. From these estimations $Q^{(ANN)}$, preliminary physically-based estima-

tions $Q^{(0)}$ based on an algebraic flow model are obtained. $Q^{(0)}$ constitutes the first guess value for the next step (VDA step).

In Section 5, the VDA method is detailed, also the ill-posedness feature of the inverse problem is discussed. In Section 6,

the physically-based estimations of $(Q(x,t), A_0(x), K(x; h(x,t)))$ obtained by VDA are presented, both for rivers inside the

learning partition and rivers outside the learning partition. Past the "calibration step" done by VDA, given newly acquired data,

real time estimations $Q^{(RT)}$ are presented in Section 7. A conclusion is proposed in Section 8. Appendix presents the rivers

geometry model, the considered Saint-Venant flow model.

## 2   Data description

### 2.1   The altimetry and in-situ data

#### 2.1.1   The different scales

Data availability is different depending on the spatial scale. Let us detail the three different scales which are considered, see

Fig. 1.

- The largest scale is the so-called "reach scale" in the SWOT scientific community, see Rodriguez and others (2012). It

   varies between a dozen of km to a few km ($\approx 5$ km), depending on the river. Here it is called the SWOT Reach Scale

   (SwReachSc).

- An intermediate scale, called Reference Data Scale (RefDataSc), corresponds to the grid employed in the reference

   models (e.g. HEC-RAS) to generate the SWOT-like data and presumed true cross-sections $A(x)$. RefDataSc is defined

   by "nodes", Fig. 1; the distance between two nodes varies between a few km to a few hundreds of meters (generally

$\approx 200$ m) depending on the river.





– The Computational Grid Scale (CompGridSc) corresponds to the computational grid of the Saint-Venant dynamics flow model (see Section B). The CompGridSc elements are 100m long.

Note that a lower complexity flow model (the algebraic model presented in Section 4) will be defined at SwReachSc only.

### 2.1.2 SWOT-like data

The future SWOT instrument will provide time series ($\sim 4 - 20$ days frequency depending on the location) of WS elevation $Z$ and water extend therefore the river width $W$, Rodriguez and others (2012); Rodriguez and Esteban-Fernandez (2010). These measurements may be available at different scales: at RefDataSc at the "nodes" location and at SwReachSc. The measured WS slopes $S$ will be accurate at large scales i.e. at SwReachSc only.

In the present study, SWOT-like data are considered as follows:

– The complete set of measurements $(Z_{r,p}, W_{r,p}, S_{r,p})$ at SwReachSc for each reach $r$ and at each instant $p$.

– The measurements of $(Z_{r,p}, W_{r,p})$ are available at RefDataSc for each "node" $r$ and at each instant $p$.

In the sequel and if ambiguous, it will be clarified at which scale the different fields and data are considered.

The SWOT instrument may provide WS measurements $(Z, W)$ at the "node scale" $200m$ long. This fine scale data is represented by data available in the Pepsi 1 and Pepsi 2 databases at RefDataSc.

Each river portion is decomposed into $R$ reaches: $r = 1, .., R$, Fig. A1. It is assumed that $(P + 1)$ instants of measurements are available; the corresponding measurements are ordered by flow elevations $Z$; the case $p = 0$ denotes the lowest water level and $p = (P + 1)$ denotes the highest.

Given a river portion, the resulting SWOT data set is $\{Z_{r,p}, W_{r,p}\}_{R,P+1}$ plus WS slope $\{S_{r,p}\}_{R,P+1}$ at SwReachSc.

Depending on the considered flow model, the $r$-th "spatial point" denotes either the node or the reach number. More precisely, the node scale is the adequate scale for the Saint-Venant dynamics flow model B1, while the larger reach scale is consistent with the low complexity algebraic model 5, see Garambois and Monnier (2015); Brisset et al. (2018) for investigations.

### 2.1.3 In-situ data

Three databases have been employed: Pepsi databases which have been built up for the Pepsi 1 and 2 challenges, see Durand et al. (2016); Frasson et al. (Submitted), and HydroSHEDS (Hydrological data and maps based on SHuttle Elevation Derivatives at multiple Scales), see Lehner et al. (2008). The Pepsi databases are a compilation of synthetic flow observations generated from outputs of various hydraulic flow models. These models have been calibrated; it is assumed that they represent the real flow dynamics. Next, SWOT like observations have been computed from these models outputs at daily sampling, both at RefDataSc (see previous paragraph) and at SwReachSc. No errors were added to the hydraulic models outputs.

The number of days, nodes and reaches varies from one river portion to another. The number of days varies from 12 days to a full year. The number of nodes by river portion varies from 21 to 3189; the number of reaches varies from 4 to 16. Some of the river portions in this dataset were outside the range of SWOT visibility since the width was less than 50m; they were


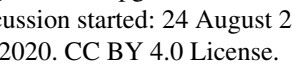


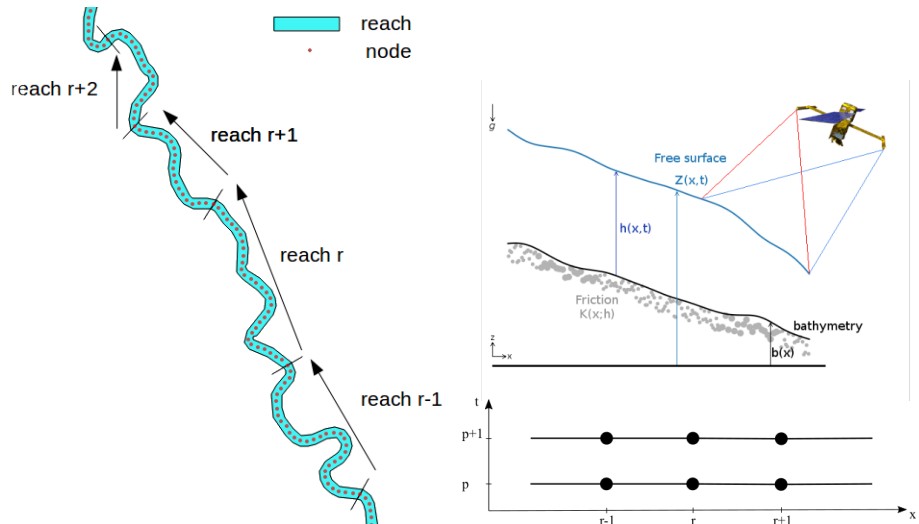

**Figure 1.** (Left) Top view of an observed river with the two different scales SwReachSc and RefDatSc.

At each each reach $r$ (cyan polygons, Swot reach) corresponds a set of WS measurements $(Z_{r,p}, W_{r,p}, S_{r,p})$. The algebraic (low complexity) flow model is solved at this scale.

At each node (red circle, RefDatSc) corresponds a presumed true cross-section $A_r$ and a set of WS measurements $(Z_{r,p}, W_{r,p})$.

At CompGridSc points (not shown here, $dx = 100$m), no SWOT like data is available. The Saint-Venant dynamics flow model is solved at this scale.

(Right)(Top) The inverse problem: infering the flow discharge $Q(x,t)$ $(m^3/s)$, the bathymetry $b(x)(m)$ (equivalently the unmeasured lowest wetted cross-section $A_0(x)$, see Fig. A1 too) and an effective friction parameter $K(x,t)$ from WS measurements $(Z,W)(x,t)$.

(Right)(Bottom) Space-time grid of the observations: reach number $r$ in $x$-axis, satellite overpass instant $p$ (re-ordered from low to high flowline) on the $y$-axis.

then removed from the dataset. Similarly river portions with less than 100 days of observations were removed. Finally, a total number of 29 river portions were selected which represents a total count of 145 reaches and (time multiplied by space) of
55 525 observations of any variable at SwReachSc. At RefDataSc, values of $(Z,W)$ and $(Q,A)$ are available. At SwReachSc, values of $(Z,W,S)$ and $(Q,A)$ are available.

HydroSHEDS is a collection of geo-referenced datasets (vector and raster) at various scales (from 3 arc seconds to 30 arc seconds). It includes river networks, void filled DEM, watershed boundaries, drainage directions and flow accumulations. As the flow accumulation in HydroSHEDS is expressed in number of cells, a dedicated script to compute drainage area (flow
accumulation in m2) from the drainage directions has been developed. Then the drainage area at every reach of the PEPSI database has been computed using the geo-location of every river portions.





## 2.2 Statistic description of the datasets

Data representing the important features of the considered rivers portions are presented in Fig. 2. More precisely for each river portion are presented the mean value, quartiles (and outliers) for the discharge $Q$ and width $W$, Fig. 2 (Top). The drainage area

$\mathcal{A}\ (km^2)$ related to the considered river portion is also plotted, Fig. 2 (Bottom). This variable is not present in the flow models however it is an important information to estimate discharges using the ANN (see next Section).

Three rivers (Jamuna, Mississipi downstream and Padma) present particularly high values of discharges and widths (as well as high values of drainage area $\mathcal{A}$). It can also be noted that the Missouri river portion presents high values of drainage area too.

This analysis helps to set up a-priori pdf and covariance kernels to solve the algebraic flow (Section 4.3) and the VDA optimization problem (Section 5.2).

The Pearson correlation coefficient $R^2$ has been computed between numerous variables: $Z, W$, elevation variations $dZ$, wetted crossed-sections variations $dA$ (both being computed between two ordered overpasses), also soil composition (percentage of clay, sand and silt), mean annual rain, mean annual temperature and land use (numerical results not presented). The only

high correlations are between: $(Q, dZ)$, $(Q, dA)$ and somehow $(dA, W)$.

## 2.3 Learning set $Q$-Lset and assessment sets ($Q$-Vset-in, $Q$-Vset-out)

The learning dataset employed in this section is constituted by river portions presenting mean discharge value lower than $10\ 000\ m^3$, see Fig. 2 (Top Left). All river portions satisfying this criteria are considered except 4 of them (which have been

randomly chosen). The resulting learning set is denoted by $Q$-Lset. It contains data related to $(24 - 4) = 20$ river portions. This corresponds to a total of $41\ 747$ "training samples" of 5 predictor variable: $(dA, W, S, \mathcal{A})$ plus one target variable $(Q)$ for each observation location and each overpass instant.

The remaining 4 rivers portions in the $Q$-Lset are: Garonne downstream, Missouri mid-section, Iowa and Ohio. They constitute the validation dataset; it is denoted by $Q$-Vset-in. $Q$-Vset-in will be used to assess the prediction capabilities of the trained

ANN (Section 3). The obtained results will show the estimation capabilities of the trained ANN to rivers presenting values of $Q$ within the learning range (hence the name $Q$-Vset-in).

In other respect, the rivers portions presenting discharge values greater than $10\ 000\ m^3$ (Jamuna, Mississippi downstream and Padma, see Fig. 2 (Top Left) are gathered in the dataset denoted by $Q$-Vset-out. $Q$-Vset-out will be used to assess the estimation capabilities of the trained ANN for rivers presenting values of $Q$ outside the learning range.

## 3 Data-driven estimations of $Q$ by ANN

In this section, purely data-driven estimations of discharge are performed and analyzed. The estimations are obtained by training an Artificial Neural Networks (ANN) from the learning set $Q$-Lset.



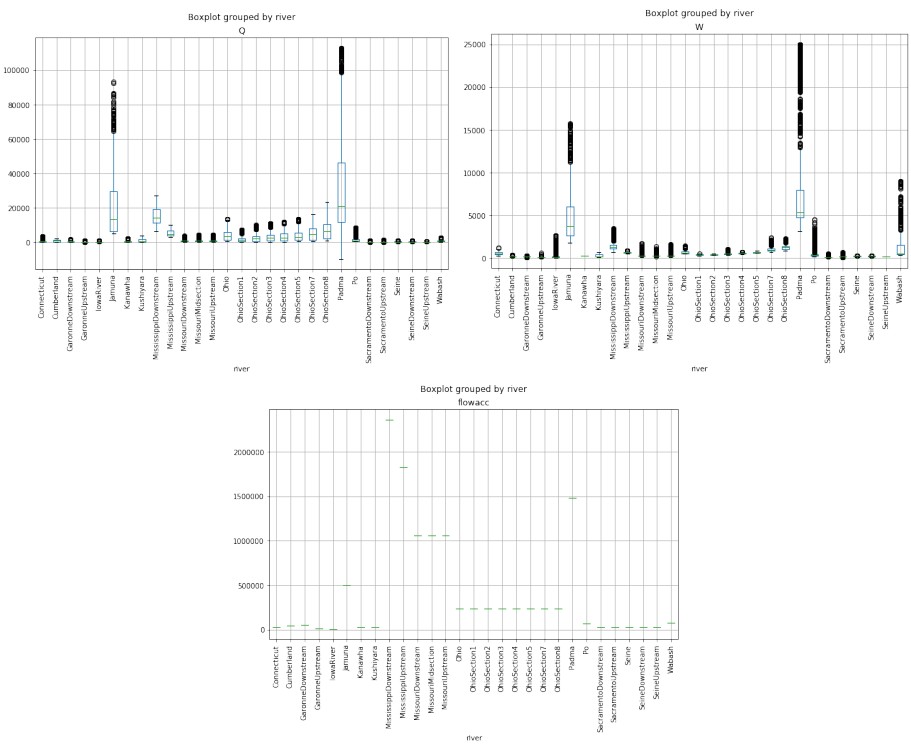

**Figure 2.** Hydraulic features of the considered river portions. The green bar indicates the mean value, boxes indicate $\pm 25\%$ quartiles, circles are outliers (Python boxplot command). (Top Left) Discharge $Q$ $(m^3/s)$ . (Top Right) Width $W$ $(m)$. (Bottom) Drainage area $\mathcal{A}$ $(km^2)$.

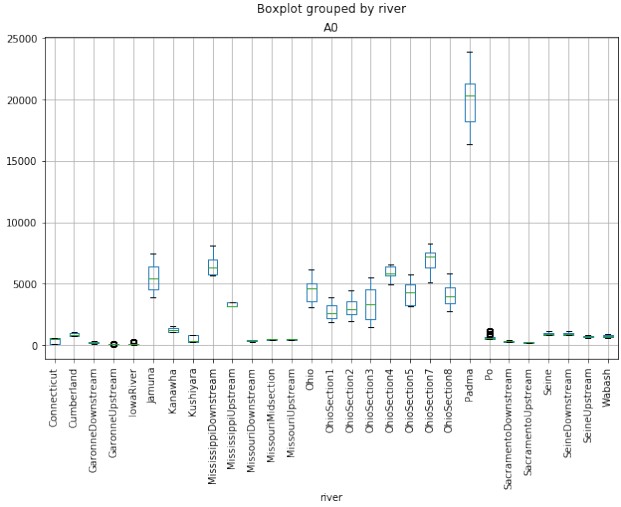

**Figure 3.** Values of the lowest wetted cross-section $A_0$. $A_0$ is an unobserved quantity.





## 3.1 The ANN description

The employed ANN is designed as follows. The training dataset $\mathcal{D}$ contains $N_{lp}$ learning pairs ("examples") $(I_i, Q_i)$, $i = $
$1, \cdots, N_{lp}$ .

The $i$-th input is $I_i = (dA, W, S, \mathcal{A})_i$ where $i$ denotes the $i$-th value at the considered location and day. (The slopes values are deduced from the WS elevation $Z$ and $dA$ implies to known $A_0$). Measurements are daily sampled.

The corresponding $i$-th output is the discharge value $Q_i$ at the same location and instant (day).

The parameters of the neural network are denoted by $W_k$, $k = 1, \cdots, N_{hl}$; $N_{hl}$ the number of hidden layers.

Each layer contains $N_{nn}$ neurons. Since neurons are connected to each other, the size of each parameter $W_k$ equals $N_{nn} \times N_{nn}$. The input variables are re-scaled by removing the mean and scaling to unit variance.

Numerous numerical experiments based on numerous different network architectures have been tested. We have observed that fairly deep networks improve the estimation capabilities (ability to find quite correctly nonlinear trends between data); From our experiments, the set $N_{hl} = 64$ and $N_{nn} = 64$ has proven the best precision w.r.t. performance.

Therefore $W_1$ contains $4 \times N_{nn} = 256$ parameters, each $W_j$, $j = 2, \cdots, (N_{hl} - 1)$, contains $N_{nn}^2 = 4096$ parameters, while $W_{N_{hl}}$ contains $N_{nn} \times 1 = 64$ parameters.

Training an ANN consists to solve the following optimization problem:

$$W^* = arg\min_{W} l_Q(W) \tag{1}$$

with the loss function (misfit-cost function) $l_Q$ classically set as

$$l_Q(W) = \frac{1}{N_{ls}} \sum_{i=1}^{N_{ls}} \left( Q_i(W) - Q^{obs}(I_i) \right)^2 = \|Q(W) - Q^{obs}(I)\|_{2, N_{ls}}^2 \tag{2}$$

(We may denote too: $Q_i^{obs} = Q^{obs}(I_i)$). The resulting estimator is:

$$Q^{(ANN)} = Q(W^*; I) \tag{3}$$

The activation function of the ANN is the usual rectified linear unit (ReLU) function, see e.g. Glorot et al. (2011); LeCun et al. (2015) for details. The ANN have been coded in Python using Keras and Mpi4Py libraries Dalcín et al. (2005). The
minimization of $l_Q(W)$ is performed using the classical Adam method Kingma and Ba (2014), a first-order gradient-based stochastic optimization. The learning rate (the gradient descent step size) is classically adjusted during the optimization procedure. As usual, the hyper-parameters of the algorithm (learning rate, decay rate, dropout probability) are experimentally chosen; the selected values are those providing the minimal value of $l_Q$. The reader may refer e.g. to Kanevski et al. (2009) for more details and know-hows on ANN algorithms. *Remark*. The drainage area $\mathcal{A}$ $(km^2)$ is not represented in the hydrodynam-
ics models, at least neither (5) nor (B1). However this information is connected to the un-modeled infiltration fluxes. In other





| Criteria | nRMSE | $R^2$ |
|---|---|---|
| Mean value for the 20 rivers | 12.85 % | 0.98 |

**Table 1.** Accuracy of the trained ANN for the 20 rivers of $Q$-Lset: obtained mean value of criteria.

words, the ANN apparently find correlations between the WS measurements, the discharge and the infiltration fluxes through the drainage area value only.

**Performance criteria**

Few criteria are used to measure the estimation accuracy: the normalized RMSE ($nRMSE$), the Nash–Sutcliffe Efficiency coefficient ($NSE$) and the Pearson correlation coefficient ($R^2$). Applied to the variable $Q$, these criteria read:

- $nRMSE(Q)$=$RMSE(Q)/\bar{Q}^{obs}$ with $RMSE(Q) = \left(\frac{1}{n}\sum_{i=1}^{n}(Q_i^{est} - Q_i^{obs})^2\right)^{1/2}$, $Q_i^{est}$ (resp. $Q_i^{obs}$) is the estimated (resp. observed) $i$-th discharge value.

- NSE criteria reads: $NSE = 1 - \frac{\sum_{i=1}^{n}(Q_i^{est}-Q_i^{obs})^2}{\sum_{i=1}^{n}(Q_i^{obs}-\bar{Q}^{obs})^2}$. NSE value range within $[-\infty, 1]$.

- $R^2$ criteria reads: $R^2(Q) = \frac{\sum_{i=1}^{n}(Q_i^{est}-\bar{Q}^{est})(Q_i^{obs}-\bar{Q}^{obs})}{\left(\sum_{i=1}^{n}(Q_i^{est}-\bar{Q}^{est})^2\right)^{1/2}\left(\sum_{i=1}^{n}(Q_i^{obs}-\bar{Q}^{obs})^2\right)^{1/2}}$ .

**3.1.1**

**Convergence and estimations for learned river portions**

After optimization (learning stage), the loss function value (2) is low: the mean values of the misfit equals 189 ($m^3/s$). The mean $nRMSE$ and $R^2$ over the 20 learned rivers are excellent, see Tab. 1. As a consequence the trained ANN is an excellent

estimator for learned rivers. The estimated discharges for the 4 first rivers in alphabetical order are presented in Fig. 4.

Let us recall that uncertainty error on discharge measurements may be considered as $\approx 30\%$ (see e.g. Gore and Banning (2017) and references therein) that is higher than the obtained nRMSE on the estimations (Tab. 1).

**3.2 Estimations for river portions within the learning partition**

Below are presented the results obtained for the 4 river portions belonging to $Q$-Vset-in, that is rivers not belonging to the

learning set $Q$-Lset but presenting mean discharge values lower than 10 000 $m^3$. These results are sort of K-fold cross-validations but applied to these 4 river portions only. These 4 river portions have been randomly chosen. The trained ANN is globally an excellent estimator for non-learned rivers belonging to the learning partition, see Tab. 2 and the hydrographs presented in Fig. 5. Garonne downstream, Missouri mid-section and Ohio hydrographs are very well estimated; the nRMSE are lower than 30%. Only the local peaks are not well captured. Note that in the Ohio case, the peak values are greater

than 10 000 ($m^3/s$), that is outside the learning range. For the Iowa, the estimated hydrograph is also accurate, excepted for the lowest values. The Iowa low flows present very low discharges which are not well estimated by the ANN; therefore the

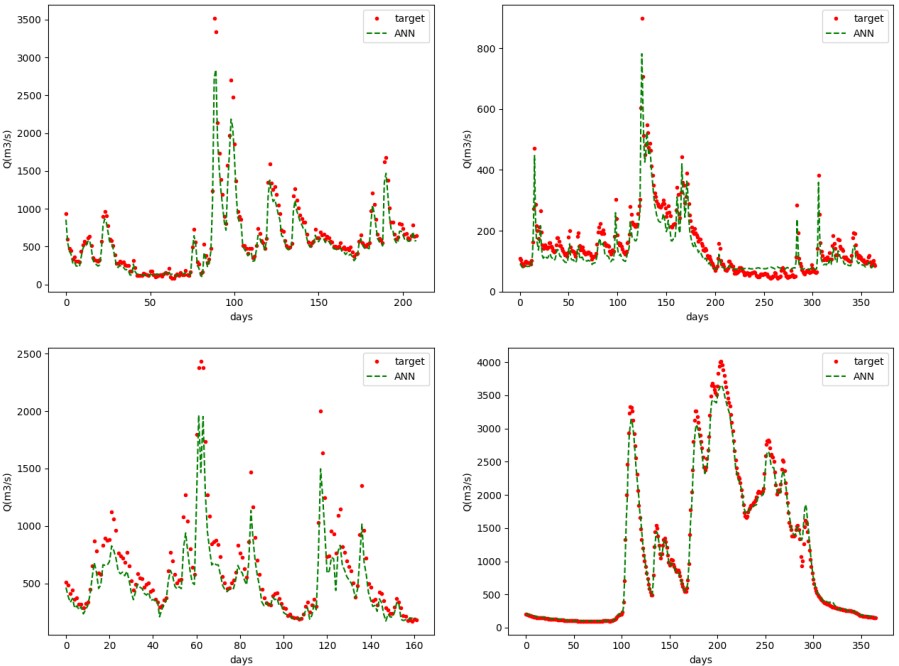

**Figure 4.** Discharge values estimated by the trained ANN for the 4 first rivers in alphabetical order belonging to $Q$-Lset. (Top Left) Connecticut. (Top Right) Garonne upstream. (Bottom Left) Kanawha. (Bottom Right) Kushiyara.

| Rivers | nRMSE | NSE |
|---|---|---|
| Garonne downstream | 26.6 % | 0.81 |
| Iowa | 44.4 % | 0.85 |
| Missouri mid-section | 18.7 % | 0.85 |
| Ohio | 18.9 % | 0.94 |

**Table 2.** Accuracy of the trained ANN for the 4 rivers of $Q$-Vset-in.

relatively high nRMSE, Tab.2. In conclusion, the present simple ANN enables to accurately estimate discharge values for rivers belonging to the learning partition.

## 3.3 Estimations for river portions outside the learning partition

Below are presented the results obtained for the 3 river portions belonging to $Q$-Vset-out that is presenting a great majority of discharge values greater than $10\ 000\ m^3$, see Fig. 2 (Top Left). The performance criteria are indicated in Tab. 3; the hydrographs are presented in Fig. 6. In all cases, the estimated values are greatly lower than the target ones. Rough variations of the hydrographs are partly recovered, although the peaks are greatly smoothed. Note that the target discharge values are up



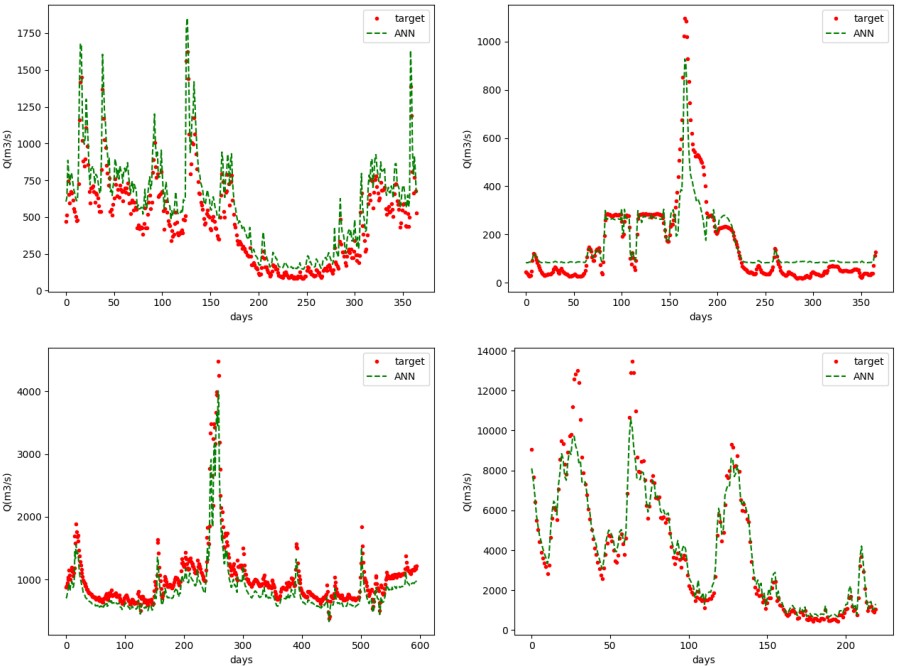

**Figure 5.** Discharge values estimated by the trained ANN for the river portions belonging to $Q$-Vset-in. (Top Left) Garonne Downstream. (Top Right) Iowa. (Bottom Left) Missouri mid-section. (Bottom Right) Ohio.

| Rivers | nRMSE | NSE |
|---|---|---|
| Jamuna | 73.3 % | 0.28 |
| Mississipi downstream | 43.6 % | -0.56 |
| Padma | 109.4 % | -0.67 |

**Table 3.** Accuracy of the trained ANN for the 3 rivers of $Q$-Vset-out.

to 10 times the discharge values considered for the learning stage. In conclusion, as expected, a purely ANN estimation does
not provide accurate estimation for rivers outside the learning partition. However, these first ANN estimations will provide in
the sequel an interesting prior.

## 4  Physically-based estimations using the algebraic flow model: first guesses

In this section, the low Froude flow model is presented; it is an algebraic system. The Strickler friction coefficient $K$ has to
depend on space and time, then to reduce its complexity, it is modeled as a power-law in water depth $h$. Next given the WS
measurements and $Q^{(ANN)}$, the algebraic flow model is solved to obtain estimations of $(A_{0,r}, (\alpha, \beta)_r)$ and $Q_{in,p} \, \forall r, p$. These
estimations will be next considered as the first guess values in the VDA based inversion presented in next section.



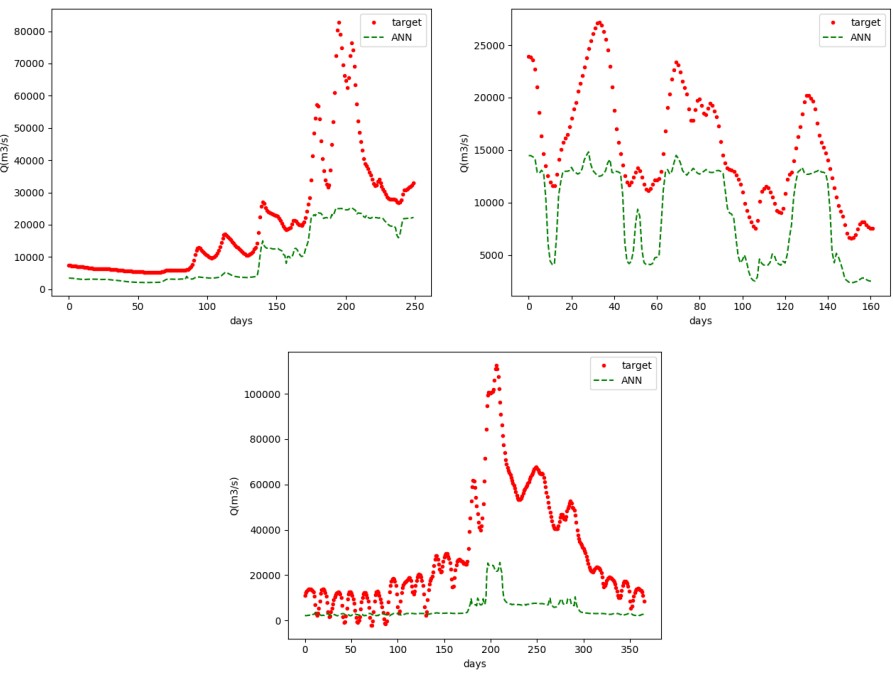

**Figure 6.** Discharge values estimated by the trained ANN for river portions belonging to $Q$-Vset-out that is rivers presenting discharge values outside the learning partition. (Top Left) Jamuna. (Top Right) Mississippi downstream. (Bottom) Padma.

## 4.1 Reduced parametrization of $K$

Following Garambois et al. (2020); Larnier et al. (2020a), the Strickler friction coefficient $K$ is defined as local power-laws at SwReachSc: $K_{r,p} \equiv K((\alpha_r, \beta_r); h_{r,p}) = \alpha_r (h_{r,p})^{\beta_r} \quad \forall r \, \forall p$. As a consequence, given $R \times (P+1)$ measurements $Z_{r,p}$, the friction parameter $K_{r,p}$ is represented by $2R$ parameters only: $(\alpha_r, \beta_r)_{1 \leq r \leq R}$. This reduced parametrization provides a local effective power-law in $h$. The law reads in function of the WS measurements as:

$$K_{r,p} \equiv K((\alpha_r, \beta_r); A_{0,r}, W_{r,0}, Z_{r,p}) = \alpha_r \left( Z_{r,p} - Z_{r,0} + \frac{1}{W_{r,0}} A_{0,r} \right)^{\beta_r} \quad \forall r \, \forall p \tag{4}$$

In the sequel if one refers to the friction parameter $K_{r,p}$, this actually refers to its parametrization defined by (4).

## 4.2 The algebraic flow model

While deriving the flow equations (mass and momentum conservation laws), the Low Froude assumption ($Fr^2 << 1$) is applied. The resulting model is an algebraic system of $R$ equations (one equation per reach $r$); each equation is similar to the





Manning-Strickler law, see Larnier et al. (2020a); Brisset et al. (2018). Since this "Low Froude" flow model is algebraic, its

complexity is low. Using the present reduced parametrization (4), this system reads as follows:

$$Q_{r,p}^{\frac{3}{5}} = \alpha_r^{3/5} \left( c_{r,p}^{(1)} A_{0,r} + c_{r,p}^{(2)} \right) \left( c_r^{(4)} A_{0,r} + c_{r,p}^{(3)} \right)^{3/5\beta_r} \quad 1 \le r \le R, \quad 0 \le p \le P \tag{5}$$

The coefficients $c_{r,p}^{(k)}$, $k = 1, \cdots, 3$, and $c_r^{(4)}$ can be evaluated from the altimetry measurements. Their expressions are:

$$c_{r,p}^{(1)} = W_{r,p}^{\frac{-2}{5}} S_{r,p}^{3/10}, \quad c_{r,p}^{(2)} = c_{r,p}^{(1)} \delta A_{r,p}, \quad c_{r,p}^{(3)} = (Z_{r,p} - Z_{r,0}), \quad c_r^{(4)} = \frac{1}{W_{r,0}} \tag{6}$$

System (5) constitutes the so-called algebraic flow model. It contains $R(P+1)$ equations.

If considering the full set of unknowns $((\alpha_r, \beta_r), A_{0,r}, Q_{r,p})$ i.e. $R(3+(P+1))$ unknowns, it is an underdetermined system therefore admitting an infinity of solutions.

If the discharge values $Q_{r,p}$ are given, the system admits an unique solution for the two other variables $((\alpha_r, \beta_r), A_{0,r})$ ($2R$ unknowns). This is the way the first guesses $(K_{r,p}, A_{0,r})^{(0)}$ are computed given $Q_{r,p}^{ANN}$, see Section 4.3.

Moreover this system will be employed differently to compute real-time estimations of $Q$, see Section 7.

Finally it is worth to notice that if $A_{0,r}$ is given $\forall r$ (therefore all wetted areas $A_{r,p} = A_{r,0} + \delta A_{r,p} \ \forall r \forall p$ are given) then by solving the algebraic flow model (5) the inference of the *ratio* $(Q/K)_{r,p}$ is possible but not the sough variables $(Q_{r,p}, K_{r,p})$. (Of course, this remark applies to the classical scalar Manning-Strickler's law too).

**Effective low Froude flow Strickler values**

Given the datasets presented in Section 2.2, the friction coefficient $K$ corresponding to the low Froude flow model is computed

by solving (5), see Fig. 7. This is an effective low Froude Strickler coefficient. This plot highlights the large range value of the effective low Froude Strickler coefficient; also it confirms physically-consistent values of $K$ obtained from the other measurements.

### 4.3    First guesses $(A_{0,r}, (\alpha, \beta)_r)^{(0)}$ and $Q_{in,p}^{(0)}$

In next section the VDA formulation is presented. It aims at estimating the unknown "input parameters" of the Saint-Venant

flow model which are: the time-dependent discharge at inflow $Q_{in}(t)$, the bathymetry $b(x)$ (equivalent to $A_0(x)$) and the friction coefficient $K$ (parametrized as $K(h(x,t)$, see (4)). The VDA algorithm is iterative; the choice of a good first guess is important. Below is presented how the first guess values $(A_{0,r}, (\alpha, \beta)_r)^{(0)}$ and $Q_{in,p}^{(0)}$ are computed.

### 4.3.1    First guess $(A_{0,r}, (\alpha, \beta)_r)^{(0)}$

Given the WS measurements and $Q^{(ANN)}$ (the discharge estimation obtained by ANN, see (3)), values of $(A_0(x), K)$ are

estimated by solving the algebraic flow model 5. These values provide the first guesses values $(A_0^{(0)}(x), K^{(0)}(h(x,t)))$ in the

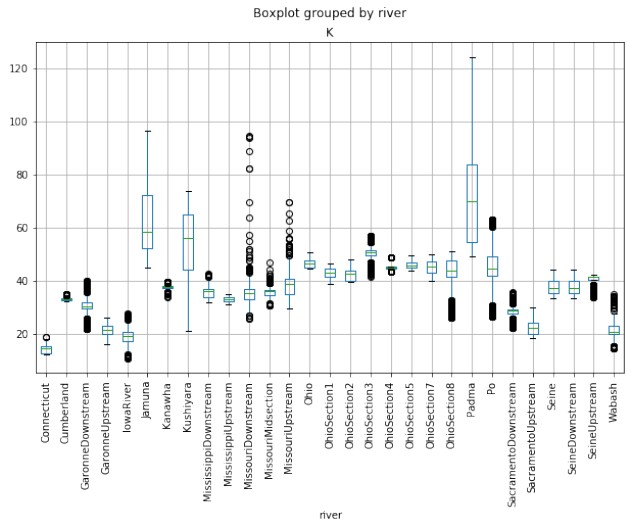

**Figure 7.** Effective Strickler friction coefficient $K$ computed by solving the low Froude (algebraic) flow model (5), given data of the considered river portions.

VDA algorithm. Recall that the Strickler friction coefficient $K$ is space-time dependent through the reduced parametrization (4). Given $Q_{r,p}^{ANN}$ (discharge estimation for reach $r$ at instant $p$), the algebraic system is solved by using the Metropolis-Hasting algorithm (MCMC method) to obtain $(A_{0,r}, (\alpha_r, \beta_r))^{(0)}$.

In the Metropolis-Hasting algorithm, the a-priori pdf are as follows: $\mathcal{U}(10, 100)$ for $\alpha_r$, $\mathcal{N}(0, 0.3)$ for $\beta_r$ and $\mathcal{N}(\mu_{A0/\bar{A}}, \sigma_{A0/\bar{A}})$
for $(A_0/\bar{A})_r$. Following the statistics obtained from the HydroSWOT and Pepsi databases $\mu_{A0/\bar{A}} = 0.73$ , $\sigma_{A0/\bar{A}} = 0.21$.

Given $A_{0,r}^{(0)}$ and the measurements $(Z_{r,0}, W_{r,0})$, the corresponding bathymetry profile $b_r^{(0)}$ is explicitly obtained, see Section 2.1. These bathymetry values are the "prior" plotted in figures 12 and 11. Note that the true values of bathymetry $b$ and $A_0$ are available at Reference Data Scale only (see Section 2).

### 4.3.2  First guess $Q_{in,p}^{(0)}$

Given $(A_{0,r}, (\alpha_r, \beta_r))^{(0)}$ , the first guess $Q_{in,p}^{(0)}$ is explicitly obtained from the algebraic flow model (5). First guess values $Q_{in,p}^{(0)}$ are plotted in Fig. 12 ("prior" curve) for rivers within the learning partition and in Fig. 11 ("prior" curve) for rivers outside the learning partition. In both cases, these low Froude estimations $Q_{in,p}^{(0)}$ catch better the variations of the true values than $Q^{(ANN)}$ (indicated as "ANN" on the figures). The estimation $Q_{in,p}^{(0)}$ may be viewed as a physically-consistent correction of the purely data driven estimation $Q^{(ANN)}$. *Remark*. If a mean value of $Q^{(true)}$ is known for a given period (e.g. a week, a
month), then one can make fit this information with $Q_{in,p}^{(0)}$. Then, with such an information $Q_{in,p}^{(0)}$ would already be an excellent estimation. However in ungauged rivers, such mean value is unavailable.





## 5 Physically-based estimations of $(Q(x,t), A_0(x), K(x; h(x,t)))$ by Variational Data Assimilation (VDA)

VDA aims at estimating the unknown "input parameters" of the Saint-Venant flow model which are the time-dependent dis-
charge at inflow $Q_{in}(t)$, the bathymetry $b(x)$ (equivalently $A_0(x)$) and the friction coefficient $K$ ($K$ is parametrized as in-
dicated in (4)). The data (WS measurements) are employed as follows. The elevation values $Z$ are used in the cost function
which measures the misfit, see (8), $W$ is used to build up the efficient cross-sections geometry of the Saint-Venant flow model,
see Section A, while the slope values $S$ are used in the algebraic flow model only, see (5).

### 5.1 The VDA formulation

The employed VDA formulation is those developed in Larnier et al. (2020a) with few improvements. At the observational
scale, the discrete unknown "parameter" of the dynamic flow model (Saint-Venant's equations) reads:

$$c = (Q_{in,0}, ..., Q_{in,P}; b_1, ..., b_R; (\alpha_1, \beta_1), ..., (\alpha_R, \beta_R))^T \tag{7}$$

The subscript $p$ denotes the instant, $p \in [0..P]$ , $r$ denotes the reach number, $r \in [1..R]$ , see Fig. A1. The parameters used
to impose a normal depth at downstream, see Section B, are considered as unknown parameters too (otherwise the flow would
be controlled by the imposed outflow condition).

The cost function aims at measuring the misfit between data (therefore at observational scale) and the Saint-Venant (fine
scale) flow model output. It is defined as:

$$j(c) \equiv j_{obs}(c) = \frac{1}{2} \sum_{p=0}^{P} \sum_{r=1}^{R} \left( Z_{r,p}(c) - Z_{r,p}^{obs} \right)^2 \tag{8}$$

This cost function $j$ has to be minimized, starting from a first guess value (prior) $c^{(0)}$. However following Lorenc et al.,
2000; Larnier et al., 2020a, the following change of variable is applied:

$$k = B^{-1/2}(c - c^{prior}) \tag{9}$$

with $B$ a covariance (symmetric definite positive) matrix, $B = B^{1/2}B^{1/2}$.

Then by setting $J(k) = j(c)$, the considered optimization problem reads:

$$\min_k J(k) \tag{10}$$

The first order optimality condition of this optimization problem reads: $B^{1/2}\nabla j(c) = 0$. The change of variable based on the
covariance matrix $B$ acts as a preconditioning of the optimization problem, see e.g. Haben et al., 2011a, b for related analysis.



Recall that in the linear-quadratic case (the model is linear, the functional is quadratic), one can show the equivalence between the VDA solution of (10) (considering (9)) and Bayesian estimations based on $B$, see e.g. Monnier (2018). The VDA algorithm is implemented in the DassFlow computational code Larnier et al. (2020b); it employs the automatic differentiation tool Tapenade Hascoët and Pascual (2013).

It is necessary to add a regularization ("convexifying") term to the cost function $j(c)$ to define a better conditioned optimization problem, see e.g. Bouttier and Courtier (2002). The classical way to do it is to define $j$ as follows: $j(c) = j_{obs}(c) + j_{reg}(c)$ with $j_{reg}$ a Tikhonov regularization term. Here the regularization term reads as: $j_{reg}(c) = \frac{1}{2}\left(\gamma_b \sum_{r=1}^{R}|\partial_r b_r(c)|^2 + \gamma_\alpha \sum_{r=1}^{R}|\partial_r \alpha_r(c)|^2\right)$.

The regularization term weight coefficients $\gamma_\square$ are empirically set (making initially the regularization terms $\approx 10\%$ of $j_{obs}$). Following an adaptive regularization strategy, see e.g. Kaltenbacher et al. (2008), the weight coefficients are divided by 2 every 10 descent algorithm iterations.

Moreover thanks to the formulation (9), a regularization term is implicitly introduced through the covariance matrix $B$ too. Indeed one can show the equivalence between the chosen covariance kernel $B$ (e.g. as the second order auto-regressive kernel like those employed below, see (12)) and a regularization functional. The reader may refer to e.g. Tarantola (2005); Monnier and Zhu (2019) for detailed examples. The definition of $B$ is detailed subsection. *Remark.* Compared to the HiVDI algorithm presented in Larnier et al. (2020a) (and implemented in Larnier et al. (2020b)), a technical but important improvement have been introduced. The vertical discretization of the river geometry (superimposition of the measured trapeziums, see Section A) is now represented by a smooth curve parametrized by a very low number of points (eg. 5); these points being optimal in the sense they minimize the $R^2$ (Pierson) criteria. Defining a regularized vertical geometry is important since it is differentiated in the reverse code. Indeed the adjoint method (implemented using automatic differentiation) computes the differential of the geometry function. Therefore if this function presents numerous stiff local gradients, this may affect the algorithm convergence robustness. The present regularized geometry provides more robust convergence of the optimizer while it is remains physically-consistent.

## 5.2 Setting the covariance matrix $B$

The choice of $B$ greatly determines the computed solution of the inverse problem; this "prior model $B$" constitutes an important feature of the VDA formulation. In the present study, these covariances are defined from classical operators but with non constant coefficients therefore defining somehow physically-adaptive regularizations.





### 5.2.1 Expression of $B$

Here the three unknown parameters $(Q_{in}(t), b(x), K(x))$ are supposed to be independent variables. This assumption is a-priori incorrect but one don't known a-priori universal covariances between these variables. As a consequence $B$ is defined as a block diagonal matrix:

$$B = blockdiag(B_{Qin}, B_b, B_K) \tag{11}$$

Each block matrix $B_\square$ is defined as a covariance matrix (symmetric positive definite matrix). The matrices $B_Q$ and $B_b$ are

set as the classical second order auto-regressive correlation matrices:

$$(B_{Qin})_{i,j} = (\sigma_{Qin}(t))^2 \exp\left(-\frac{|t_j - t_i|}{T_{Qin}}\right) \text{ and } (B_b)_{i,j} = (\sigma_b(x))^2 \exp\left(-\frac{|x_j - x_i|}{L_b}\right) \tag{12}$$

The matrix $B_K$ is set as $B_K = blockdiag(B_\alpha, B_\beta)$ with:

$$(B_\alpha)_{i,j} = \sigma_\alpha^2(x) \exp\left(-\frac{|x_j - x_i|}{L_K}\right) \text{ and } B_\beta = diag(\sigma_\beta^2(x)) \tag{13}$$

The parameters $T_{Qin}$ and $(L_b, L_K)$ act as correlation lengths.

### 5.2.2 Setting of the parameters $\sigma_\square$ and $(T_{Qin}, L_\square)$

These parameters are important prior information of the inversions. They are set from the first guesses values $(Q_{in,p}, A_{0,r}, (\alpha, \beta)_r)^{(0)}$.

Recall that the observation frequency is 24h. The measurements spacing varies from a few dozen meters to a few hundreds of meters. Local Froude numbers range in great majority within $\approx [0.05 - 0.3]$, with some very local maximum values up to $\approx 0.5$.

The discharge parameters are set as follows. $T_{Qin} = 24$ h. The normalization coefficient $\sigma_{Qin}$ is time-dependent: $\sigma_{Qin}(t)$ equals 30% of the mean value of $Q^{(0)}(t)$. (Recall that uncertainty error on discharge measurements may be considered as $\approx 30\%$, see e.g. Gore and Banning (2017) and references therein).

Concerning the bathymetry, $\sigma_b$ is space dependent: $\sigma_b(x)$ is set such that it corresponds to $P_{\sigma_b} = 50\%$ of the mean value of $A_0^{(0)}(x)$. Recall that the bathymetry values $b^{(0)}(x)$ are deduced from the unobserved flow area values $A_0^{(0)}(x)$.

Concerning the correlation length, we set: $L_b = 1$ km. However if this last parameter is too large, the matrix $B_b$ may be not positive. In such a case, the characteristic length $L_b$ is adaptively decrease until the matrix becomes positive. This has happened in a few cases, then the minimal resulting value was $L_b = 500$ m.

The normalization coefficients related to the friction are constant: $\sigma_\alpha = 10$ and $\sigma_\beta = 0.3$. These values have been chosen following statistical analysis made on the databases and by analyzes on the gradient components. We set $L_\alpha = dx = 100$m ($dx$

is the computation grid spacing).





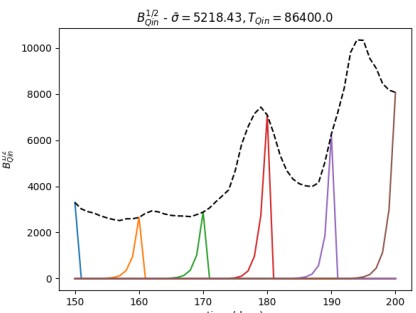
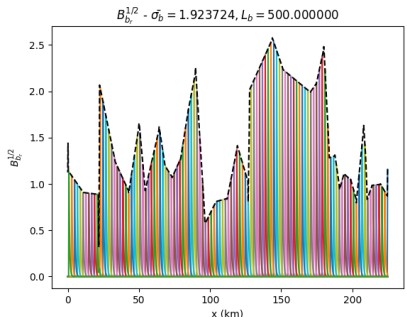

**Figure 8.** The covariance matrices $B_\square^{1/2}$ in the Jamuna river case. (L) $B_{Qin}^{1/2}$ (with $T_{Q_{in}} = 24h$). A few covariance values only are plotted for sake of readability. (R) $B_b^{1/2}$ (with $L_b = 500$m). Note that the scaling factor of $B_\square^{-1/2}$ is $\sigma_\square^{-1/2}$ and not $\sigma_\square$.

As an illustration, some covariance matrices $B_\square$ are plotted in Fig. 8.

## 5.3 Capabilities and limitations of the inversions based on the flow models only

The (low-complexity) algebraic flow model (based on the Low-Froude assumption $Fr^2 << 1$, see Garambois and Monnier (2015); Brisset et al. (2018)), enables to determine the ratio $Q/\alpha$, equivalently $Q/K$, and not the unknowns pair $(Q, K)$. This remarks holds even if the bathymetry $b$ (equivalently $A_0$) is known or not.

Let us show that the inverse problem aiming at estimating the triplet $(Q_{in}(t), A_0(x), K(h(x,t))$ in the Saint-Venant system (B1) is ill-posed too in the following sense: the model solution $(A, Q)(x,t)$ is unchanged if multiplying these unknown
parameters by an adequate multiplicative factor.

Let $\bar{Q}$ be any scalar value: $\bar{Q}$ may be a mean value of $Q$ or $K$. Let us define the following re-scaled state variables: $(A_*, Q_*) = (A, Q)/\bar{Q}$. The mass equation (B1)(a) divided by $\bar{Q}$ is unchanged therefore: $\partial_t(A_*) + \partial_x(Q_*) = 0$. The mass equation still holds; $Q$ simply implies to rescale $A$ by the same factor.

The re-scaled momentum equation ((B1)(b) divided by $\bar{Q}$) reads:

$$\partial_t(Q_*) + \partial_x\left(\frac{Q_*^2}{A_*}\right) + gA_* \partial_x Z = -gA_* S_f \tag{14}$$

with $S_f \equiv S_f(A, Q; h; K) = \frac{1}{K^2}\frac{|Q|Q}{A^2 h^{4/3}}$. If defining $h$ as the effective cross-section depth: $h = A/W$, $W$ the WS width, then: $S_f(A, Q, h; K) = S_f(A_*, Q_*, h_*; \bar{Q}^{-2/3} K)$.

Therefore, given the WS measurements $(W, Z)$, the 1D Saint-Venant equations (B1) with parameter $K$ are equivalent to the same equations in the re-scaled variables $(A_*, Q_*)$ but with $(\bar{Q}^{-2/3} K)$ as Manning-Strickler's parameter. Concerning boundary
conditions, both upstream and downstream conditions are transparently re-scaled by the factor $\bar{Q}$. This little calculation has



been first presented in Larnier et al. (2020a). A consequence of this "equifinality issue" is the following: at each minimization iteration in the VDA process (see Section 5 for details), the "model constraint" (B1) is satisfied by an infinity of flow states values $(A,Q)$ characterized by the parameter $K$. In other words, the flow model (B1) constrains the inverse problem solution $(Q_{in}(t), A_0(x), K(h))$ up a to a multiplicative factor only. This feature is (of course) retrieved in the numerical results (see

Section 6.3): the space-time variations of the infered discharge values are accurate however up to a "bias". This bias depends to the prior information introduced in the VDA formulation, see Section 5.3 for details on the introduced priors. Note that a mean value (e.g. seasonal or annual) of $Q$ may be enough to answer the issue.

In the case the bathymetry $b(x)$ is given (therefore $A_0(x)$), the re-scaled unknown $(A,Q^*)$ does not satisfy the flow model anymore. In other words, if the bathymetry is given, the inverse problem based on the Saint-Venant model (B1) may be well

posed. In other respect, recall that a single measurement of bathymetry enables the bathymetry estimation along a relatively long river portion, see Garambois and Monnier (2015); Brisset et al. (2018).

In the case a mean value of $Q$ is known (e.g. seasonal or annual value) then this information enables to fix the multiplicative factor $\gamma$; the considered inverse problem may be well-posed. Actually, the numerical results show that in such a case, the estimations of $Q(x,t)$ are accurate without bias, see Section 6.3.


As a consequence, the model-constraint of the optimization problem (10) is constraining (in space-time) but up to a multiplicative factor only. The VDA solution (solution of (10) under the dynamic flow model constraint) depends on the prior: the first guess but also the covariance matrix $B$, see Section 5.2. The introduction of $B$ is necessary to make the algorithm convergence robust. However the solution has to satisfy this prior probabilistic model $B$ (defined from the prior parameters

$\sigma_\square$). Therefore, the prior parameters $\sigma_\square$ play an important role in the determination of the optimal solution. This feature of the VDA process is well known, see e.g. Lorenc (1988); Haben et al. (2011b) and references therein. (Also the reader may refer e.g. to Monnier (2018) for a formal proof showing the equivalence between VDA covariance based solutions and Bayesian estimations).

In the present approach, the first guesses $(Q_{in}(t), A_0(x), K)^{(0)}$ of the minimization algorithm are consistent with the alge-

braic flow model (5). This model (5) defines a physically-consistent solution, however up to the multiplicative factor. Next, the descent algorithm explores the optimal solution in a "vicinity" of a physically-consistent solution, the provided first guess $(Q_{in}(t), A_0(x), K)^{(0)}$. In practice, and as illustrated by the numerical results shown in next sections, the space-time variations of $Q$ are always accurately identified but the global estimation may still present a bias; the latter depending on the accuracy of the first guess value $Q_{in}^{(0)}$ in particular.

Remark that if one value of bathymetry is known (a measurement is available at one location) then the unknown multiplicative factor issue is a-priori solved. Indeed it is shown in Brisset et al. (2018); Garambois and Monnier (2015) that if a bathymetry estimation can be obtained, and following the explanation presented in Section **??**, the inverse problem may be well-posed. Also, if any mean value of $Q$ is known (on any period e.g. annual) then as highlighted in the forthcoming numerical results (Section 6.3), the multiplicative factor issue is also solved.


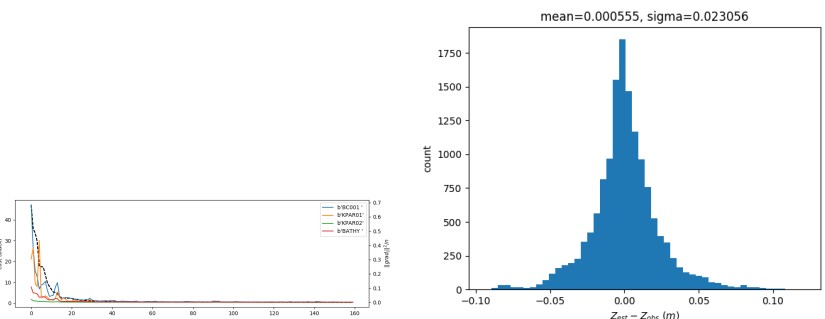

**Figure 9.** VDA algorithm convergence: (Left) A typical convergence curve: cost function $J(k)$ (dashed black line) and gradient components (colored solid lines) vs iterations (Garonne Downstream case). 'BCOO1', 'KPAR01', 'KPAR02', 'BATHY' corresponds to the gradient component wrt to $Qin(t)$, $\alpha(x)$, $\beta(x)$, $b(x)$ respectively (all in norms 2). (Right) Misfit values $|Z_{r,p} - Z_{r,p}^{obs}| \ \forall r \forall p$ in meters (see (8)) after convergence.

## 6 Estimations obtained by VDA

In this section, numerical results based on the VDA method are presented. Given a first guess $(Q_{in,p}^{(0)}, A_{0,r}, (\alpha, \beta)_r)^{(0)}$ computed as indicated in Section 4.3, the optimization problem (10) is solved by a minimization algorithm, see Section 5. First, typical behaviors and accuracy of the minimization algorithm are briefly presented. Next, numerical results are presented for two river

portions (randomly chosen) belonging to $Q$-Vset-in, rivers presenting mean discharge values within the learning range (i.e. lower than $10\,000\,m^3$) but outside the learning partition $Q$-Lset, see Section 2.3. Finally, numerical results are presented for two river portions (randomly chosen) belonging to $Q$-Vset-out, that is rivers presenting mean discharge values outside the learning range (i.e. greater than $10\,000\,m^3$), see Section 2.3.

### 6.1 On the VDA algorithm convergence

The minimization algorithm aiming at solving (10) converge generally in less than $100$ iterations; and in some complex case, the convergence may be reached after more than $150$ iterations, see e.g. Fig. 9. After convergence, the misfit values on WS elevation, see (8), is always excellent: standard deviation $\sigma_{misfit} \approx 10$ cm, see Fig. 9. This value of $\sigma_{misfit}$ is lower than the expected value for SWOT instrument ($\sigma_{SWOT} = 25$ cm, see Rodriguez and others (2012)).

### 6.2 Numerical results for rivers within the learning range but outside the learning partition $Q$-Lset

Numerical results for Garonne downstream and Ohio river portions (randomly chosen) belonging to $Q$-Vset-in are presented in Fig. 5. They are rivers presenting a mean discharge value within the learning range (i.e. lower than $10\,000\,m^3$) but outside the learning partition $Q$-Lset, see Section 2.3. As previously mentioned, the first guess values $(Q_{in,p}; A_{0,r}, (\alpha, \beta)_r)^{(0)}$ are computed by solving the algebraic flow model given the WS measurements and the prior $Q^{(ANN)}$. Performance criteria are indicated in Tab. 4; the estimations are plotted in Fig. 10 ("prior (VDA)" denotes $Q_{in,p}^{(0)}$).



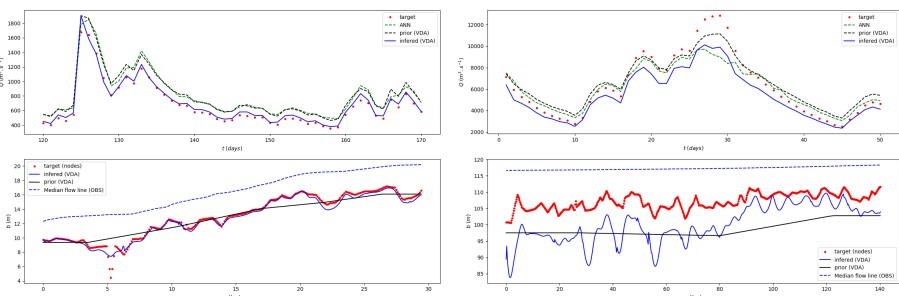

**Figure 10.** Discharge estimations for 2 rivers (randomly chosen) belonging to Q-Vset-in. (Left column) Garonne downstream. (Right column) Ohio.

(Up) Discharge values Qin(t) vs time during the assimilating period only. (Down) Bathymetry values $b(x)$ and the observed median flow line $Z_{med}^{(obs)}(x)$.

| River name | ANN prior | nRMSE | NSE |
|---|---|---|---|
| Garonne downstream | $Q^{(ANN)}$ | 28.6 % | 0.78 |
| Ohio | $Q^{(ANN)}$ | 18.5 % | 0.84 |
| Jamuna | $Q^{(ANN)}$ | 71.2 % | -0.56 |
| Mississipi downstream | $Q^{(ANN)}$ | 46.1 % | -1.00 |
| Jamuna | $2Q^{(ANN)}$ | 39.4 % | 0.52 |
| Mississipi downstream | $2Q^{(ANN)}$ | 15.0 % | 0.78 |

**Table 4.** Performance obtained on the discharge estimation $Q^{(VDA)}$ .

The three estimations (ANN, first guess and VDA solution) are excellent. Again, the nRMSE are lower than the standard error made on discharge measurements. Again, the VDA estimation (physically-based) captures better the variations than the purely data-driven estimation (ANN).

Concerning the bathymetry, in the Garonne case, the VDA clearly improve its estimation, in particular in pools (low values of $b(x)$), Fig. 10 (Down)(L). In the Ohio case, the bathymetry estimation remains relatively inaccurate despite the excellent

discharge estimation. In such a case, the bathymetry error is balanced by a Strickler coefficient adjustment.

### 6.3   Numerical results outside the learning range

Numerical results for Jamuna and Mississipi downstream portions (randomly chosen) belonging to $Q$-Vset-out are presented in Fig. 11. They are rivers presenting a mean discharge value outside the learning range (i.e. greater than $10\,000\,m^3$) therefore outside the learning partition $Q$-Lset too, see Section 2.3.




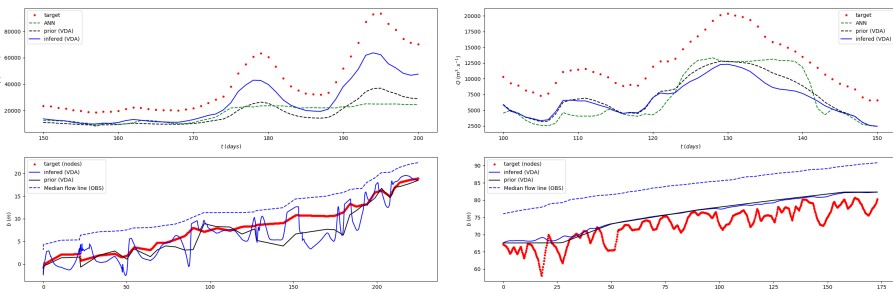

**Figure 11.** Discharge estimations for 2 rivers (randomly chosen) belonging to $Q$-Vset-out. (Left column) Jamuna. (Right column) Mississipi downstream.

(Up) Discharge values $Q_{in}(t)$ vs time during the assimilating period only. (Down) Bathymetry values $b(x)$ and the observed median flow line $Z_{med}^{(obs)}(x)$.

### 6.3.1  Estimations from the prior $Q^{(ANN)}$

In this first experiment set, the considered prior discharge value is $Q^{(ANN)}$, like in all previous numerical experiments. Recall that this prior is inaccurate since the rivers do not present mean discharge values within the learning range, see Section 3.3. The obtained estimations are presented in Fig. 11 (where "prior" denotes $Q_{in,p}^{(0)}$); performance criteria are indicated in Tab. 4.

In the Jamuna case, the VDA estimation is better than the provided first guess, that is the algebraic flow model solution ("prior VDA"in Fig. 11). This estimation captures quite well the time discharge variations; however it remains an under-estimation of the true value. The bathymetry estimation is not a real improvement of the prior value. Note that its oscillations can be easily smoothed by increasing the weight of the regularization term, see Section 5.1.

In the Mississippi case, the VDA estimation captures well the discharge variations, better than ANN again, but it deteriorates the global accuracy. In this case, the prior introduced in the covariance matrix $B$ are not satisfying. These two examples illustrate the phenomena previously mentioned: a) the flow models act as physically-consistent filters (the variations are quite well captured); b) the inverse problem is well-posed but up to a multiplicative factor only (see Section 5.3).

### 6.3.2  Estimations from a re-scaled prior

As previously discussed, the inverse problem based on the flow model without prior information is intrinsically ill-posed: one can estimate the space-time variations of the discharge values but up to a bias only, see Section 5.3. The numerical results obtained for rivers outside the learning range illustrate this feature, see Fig. 11 (see also Larnier et al. (2020a)). The knowledge of a mean value of $Q$ (e.g. annual) would solve this bias issue. To illustrate this statement, the same numerical experiments as the previous ones are conducted excepted that the prior is $2Q^{(ANN)}$ instead of $Q^{(ANN)}$. (The multiplicative factor 2 is chosen as a simple and roughly correct re-scaling; it does not correspond to any precise mean value of $Q$). The obtained discharge estimations are plotted in Fig. 12 ("prior (VDA)" denotes $Q_{in,p}^{(0)}$); performance criteria are indicated in Tab. 4.


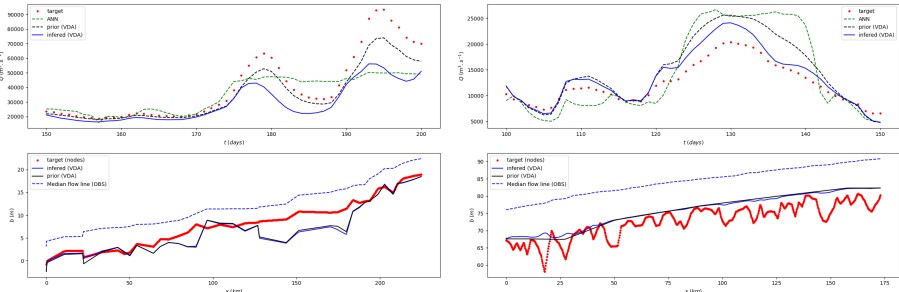

**Figure 12.** Discharge estimations for 2 rivers belonging to $Q$-Vset-out computed by VDA from the prior $Q^{(ANN)}$ multiplied by 2. Discharge values $Q_{in}(t)$ vs time during the assimilating period only. (Left column) Jamuna. (Right column) Mississipi downstream.

As expected, the global estimations are much better, see Tab. 4; also the variations are well captured by the physically-based estimations (on contrary to the purely ANN-based ones). However, surprisingly in the Jamuna case, the VDA degrades the first guess estimations ("prior (VDA)"); on the opposite in the Mississippi case, the VDA improves the first guess estimation. However in all cases, the obtained nRMSE seems to be at least among the best ones existing up to now, see Frasson et al. (Submitted).

In conclusion these results confirm that for fully ungauged rivers (without any prior information, even not a rough mean value), the estimations can present a important bias but capture accurately the time discharge variations. And if any mean value of $Q$ is known (e.g. annual) then the estimations become accurate with nRMSE very likely less than 30%.

## 7    Estimations from newly acquired data

Given a river portion, the VDA, process enable calibrating the flow model: the river has been "learned". In particular, the
estimation of an effective bathymetry $b(x)$ (equivalently $A_0(x)$) is available. Next, given newly acquired WS measurements, three discharge estimators may be employed: the trained ANN (Section 3), the algebraic flow model (5) and the calibrated flow model (B1).

If the newly acquired data belong to the learning values ranges (in terms of $(Z, W)$ values) then either the ANN or the dynamics flow model (B1) with $Q_{in}(t)$ identified by VDA may be satisfactorily employed. Note that the model (B1) is a-priori
more accurate than the ANN model since physically consistent, see Section 6.2 for an illustration of this feature. Moreover (B1) enables space-time extrapolation of discharge outside the measurements locations; this is not possible if using the ANN model. However a particularly interesting alternative is to employ the "low cost" algebraic flow model (5). This is what is illustrated in this section.

### 7.1    Estimations based on the algebraic flow model

Given newly acquired data, a strategy to estimate $Q$ in real computational time can be as follows.





– *Step 1) Recalibration of the friction coefficient $K$.*

Given $(Q_{r,p}^{(VDA)}, A_{0,r}^{(VDA)})$ obtained after the VDA process, the algebraic model (5) is solved to obtain $(\alpha_r, \beta_r)^{(LF)}$. This computation provides the effective low-Froude friction parameter $K_{r,p}^{(LF)}$; $K_{r,p}^{(LF)} = K((\alpha_r, \beta_r)^{(LF)}, A_{0,r}; Z_{r,p})$, see (4).

– *Step 2) Estimation from newly acquired data using the algebraic flow model* (5).

Given $(A_{0,r}^{(VDA)}, (\alpha_r, \beta_r)^{(LF)}) \; \forall r$, given new WS measurements $(Z_{r,p}, W_{r,p}, S_{r,p})_{R,P+1}$, the coefficients $(c_{r,p}^{(k)})$, $k = 1, 2, 3$, and $c_r^{(4)}$ in (5) can be evaluated. Then, the estimation $Q_{r,p}^{(RT)}, \forall r \; \forall p$, can be explicitly obtained from (5), therefore computed in real time.

*Remark*. No uncertainty envelope on $Q^{(RT)}$ is presented because of the necessarily arbitrary choices if doing so. Indeed, one could easily present uncertainties as follows. At Step 1), one can introduce an uncertainty model on $Q^{(VDA)}$ by considering it as a random variable e.g. $Q^{(VDA)} \sim \mathcal{N}(\bar{Q}^{(VDA)}, \sigma_Q)$. Then if using the Metropolis-Hasting algorithm to compute the effective Low-Froude values $(\alpha, \beta)_r^{(LF)}$, one obtains $K_{r,p}$ as a random variable with a corresponding standard deviation $\sigma_K$. Next at Step 2), the "Real-Time" estimation denoted by $Q_{r,p}^{(RT)}$ (explicit solution of (5)) is a random variable with a corresponding standard deviation $\sigma_{final}$. Therefore by setting a priori uncertainty on the pdf of $Q^{(VDA)}$ and $K_r$ (or $(\alpha, \beta)_r$ e.g. as in Section 4.3 with $\mathcal{U}(10, 100)$ for $\alpha_r$, $\mathcal{N}(0, 0.3)$ for $\beta_r$), one obtains the resulting uncertainty on $Q^{(RT)}$.

## 7.2 Numerical results

The VDA estimations in Section 6 have been obtained from relatively short time periods compared to a complete year, see Tab. 5; however the chosen periods are relatively representative of the potential annual variations. These "VDA periods" are the "calibration" (or "learning") periods. Outside these calibration periods, the WS measurements are considered as newly acquired; then, the real-time discharge $Q^{(RT)}$ is estimated as described in the previous paragraph. Moreover the real-time estimation $Q^{(RT)}$ is computed for the calibration period too, see Fig. 13. This enables to compare $Q^{(RT)}$ with $Q^{(target)}$ for all periods. For the two rivers outside the learning partition $Q$-Lset (Jamuna and Mississipi downstream), the prior value is $2Q^{(ANN)}$ and not $Q^{(ANN)}$. The discharge estimations are plotted in Fig. 12. In the calibration periods, the estimation $Q^{(RT)}$ differs from $Q_{in,p}^{(0)}$ ("prior VDA") because of the use of the bathymetry estimation obtained after VDA. Performance criteria are indicated in Tab. 5.

Again for the rivers belonging to the learning partition (Garonne downstream, Ohio) therefore with an already excellent estimation by ANN, all the estimations are accurate (17-28% nRMSE). For the rivers (far) outside the learning partition (Jamuna and Mississippi downstream),the nRMSE remains good $(21 - 35\%)$, equal to the order of magnitude of the errors when measuring discharges. In these two cases, the real-time estimation (which physically-based) is better than the purely data-driven prior $2Q^{(ANN)}$. However, it can be noticed that $Q^{(RT)}$ (solution of a re-run after the VDA calibration) may be better or worse than the prior VDA $Q_{in}^{(0)}$. However, again in all cases the obtained nRMSE is at least among the best ones existing up to now for ungauged rivers, Frasson et al. (Submitted); moreover these estimations are computed in real-time given newly acquired WS measurements.

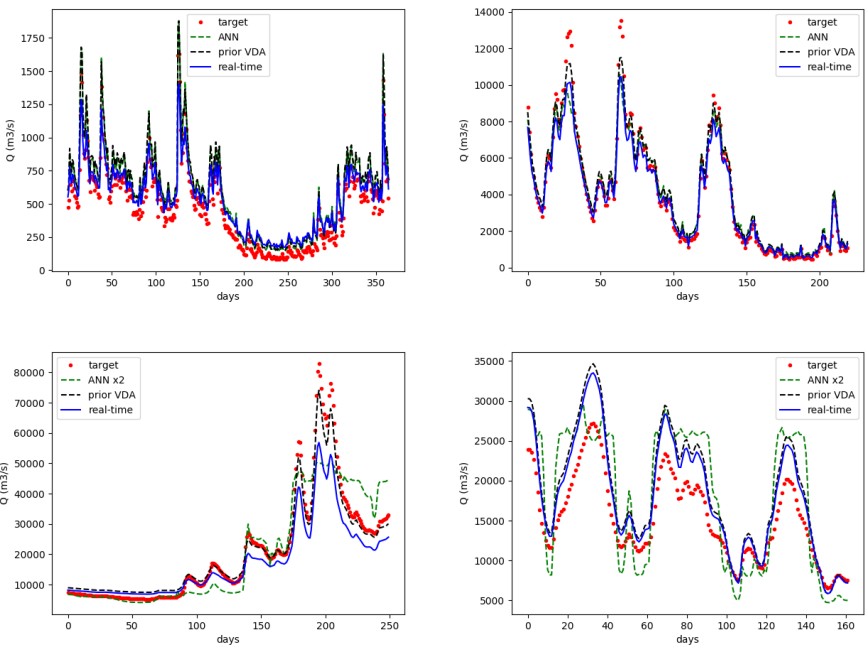

**Figure 13.** Real-time discharge estimations $Q(t)$ vs time during the complete time period by solving (5). "Prior" corresponds to the first guess $Q_{in,p}^{(0)}$ presented in Section 4.3.

(Up)(L) Garonne downstream. (Up)(R) Ohio.

(Down) (L) Jamuna. (Down)(R) Mississipi downstream.

| River name | Calibration period (days) | Complete estimation period (days) | nRMSE | NSE |
|---|---|---|---|---|
| Garonne downstream | [120-170] | [1-365] | 28.6 % | 0.78 |
| Ohio | [1-50] | [1-220] | 17.3 % | 0.95 |
| Jamuna | [150-200] | [1-250] | 35.0 % | 0.84 |
| Mississipi downstream | [100-150] | [1-162] | 21.8 % | 0.61 |

**Table 5.** The calibration periods are those considered in the VDA processes. The performance scores are those obtained for $Q^{(RT)}$ during the complete period.

## 8 Conclusions

This study proposes the first combined deep learning - data assimilation approach to infer river discharge values from altimetry measurements only (SWOT like data). The resulting algorithm, named HiVDI (Hierarchical Variational Discharge Inference), is an important improvement of the former version presented in Larnier et al. (2020a), see also Frasson et al. (Submitted). This algorithm relies on: preliminary statistical analysis of the WS measurements and drainage areas, a deep neural network





providing a first estimation of $Q$ which is next improved by a low Froude flow model (algebraic model). The resulting discharge

estimation is excellent for ungauged rivers presenting discharge values within the learning partition ($\approx 10 - 30\%$ nRMSE). For river discharges outside the learning partition, the time variations are very well captured but a bias remains; this bias is inherent to the automatic learning approach. Next, an advanced Variational Data Assimilation (VDA) method enables the estimation of accurate space-time variations of $Q(x,t)$ for any location at any time; however the potential bias may remain. It is shown that this bias cannot be removed if defining the estimations by inverting the flow models only. However a simple mean value

of $Q$ (eg. seasonal or annual) enables to remove the bias, therefore providing accurate estimations ($\approx 30\%$ nRMSE). In short, the estimation is based on automatic learning and the inversion of two hierarchical flow models. Given a representative WS measurements set (eg. during a complete year), the method results on two calibrated flow models of hierarchical complexity (algebraic and dynamics) for each river portion. Next, given newly acquired WS measurements, the algebraic flow model is accurate enough and low-cost enough to provide in real time the estimation of $Q$ at km scale and at the hours observation scale.

The HiVDI algorithm is implemented into the open-source computational software DassFlow Larnier et al. (2020b). It may be used for an operational purpose when the SWOT instrument will be launched in 2021.

**Appendix A: River geometries**

Recall that the SWOT-like measurements consist in sets $(Z_{r,p}, W_{r,p})_{R,P+1}$ . Moreover at SwReachSc, WS slopes values $S_{r,p}$ are available and taken into account into the algebraic flow model (see next section). The values $S_{r,p}$ are either deduced from

the elevation values $Z$ or estimated by an internal instrument process.

The considered river geometries are derived from the datasets $(Z_{r,p}, W_{r,p})_{R,P+1}$. The cross-sectional geometry consists in discrete cross sections formed by asymmetrical trapezium layers $(Z_{r,p}, W_{r,p})$, see Fig. A1. The cross-sectional areas $A_{r,p}$ satisfy: $A_{r,p} = A_{r,0} + \delta A_{r,p} = A_{r,0} + \int_{Z_{r,0}}^{Z_{r,p}} W_r(h)dh \ \ \forall r \ \forall p \geq 1$.

The variations $\delta A_{r,p}$ are approximated by the trapeziums: $\delta A_{r,p} \approx \sum_{q=1}^{p} \frac{1}{2}(W_r^q + W_r^{q-1})(h_r^q - h_r^{q-1})$.

The lowest cross-sectional areas denoted by $A_{r,0}$ ($r = 1, \cdots, R$) are unobserved; they are key unknowns of the flow models. They can be represented by rectangles or any other fixed shape (e.g. a parabola); all the other cross-sectional areas are trapezoidal. Next, for simplicity and regularization purposes, the shape is approximated at a cubic spline curve in the least square sense (green curve), see Fig. A1.

For all considered rivers we have the hydraulic radius $R^h$ which satisfies: $R_{r,p}^h \approx h_{r,p}$. Also since $W >> h$, it follows the

effective depth expression: $h_{r,p} \approx (A_{r,0} + \delta A_{r,p})(W_{r,0} + W_{r,p})^{-1}$.

**Appendix B: The dynamic flow model**

The considered dynamic flow model is the 1D Saint-Venant equations in their non conservative form in $(A, Q)$ variables; $A$ the wetted-cross section $\left[m^2\right]$, $Q$ the discharge $\left[m^3 . s^{-1}\right]$. The equations read as follows, see e.g. (Chow, 1964):

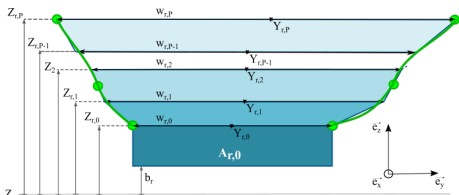

**Figure A1.** A cross section $A_r$ is the superimposition of the observed trapeziums $A_{r,p}$ defined from the $p$-ordered SWOT measurements $(Z_{r,p}, W_{r,p})$ + the unobserved lowest cross-section $A_0$. Next, the shape is approximated by a cubic spline (green curve).

$$
\quad \begin{cases} \partial_t A + \partial_x Q & = 0 \\ \partial_t Q + \partial_x \left( \frac{Q^2}{A} \right) + gA\,\partial_x Z & = -g\,A\,S_f(A, Q; K) \end{cases} \tag{B1}
$$

where $g$ is the gravity magnitude $\left[ m.s^{-2} \right]$, $Z$ is the WS elevation $[m]$, $Z = (b + h)$ where $b$ is the lowest rectangular cross-section (bed) level $[m]$ and $h$ is the water depth $[m]$.

At inflow (upstream), the discharge $Q_{in}(t)$ is imposed.

At outflow (downstream), if known the WS elevation $Z_{out}$ is imposed. If unknown, the normal depth (based on the Manning-Strickler equilibrium equation) is imposed. Recall that the normal depth depends on the prior values $(K, A_0)$ at outflow.

The RHS term $S_f$ is the classical Manning-Strickler friction term: $S_f(A, Q; K) = \frac{|Q|Q}{K^2 A^2 R_h^{4/3}}$ with $K$ the Strickler roughness coefficient $\left[ m^{1/3}.s^{-1} \right]$ with $R_h \approx h\ [m]$. $K$ is defined following the local power-law (4). The discharge $Q$ is related to the water velocity $u\ \left[ m.s^{-1} \right]$ by the relation: $Q = uA$.

This 1D Saint-Venant model is discretized using the classical implicit Preissmann scheme (see e.g. Cunge, 1980) with a space cell length $\Delta x = 200$m and time step $\Delta t = 1$h.

The numerical model has been implemented in the computational software DassFlow Larnier et al. (2020b).

*Author contributions.* J.M. has designed the research plan, the methods, equations and algorithms; he has written the manuscript. K.L. has implemented the algorithms and has performed the numerical results. Both authors have contributed to the results analysis and the methods
selection.

*Acknowledgements.* K. Larnier, software engineer at CS group corp., has been funded by CNES. The authors would like to acknowledge Miss Ha Nhi Ngo, INSA 5th year student, which has provided a preliminary Jupyter - Python sheet facilitating the statistical analysis. The authors would like to acknowledge Renato Frasson (NASA/JPL) and Mike Durand (Univ. Ohio) for sharing Pepsi-2 dataset.





*Competing interests.* No competing interests are present.



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
