# Peer review of "Hybrid Neural Network - Variational Data Assimilation algorithm to infer river discharges from SWOT-like data"

_Nonlinear Processes in Geophysics, 2020_

## Referee Comment (RC1) · Anonymous Referee #1 · 3 Sep 2020

In this paper, the authors developed a new method for river bathymetry and discharge estimation from satellite altimetry data. They firstly estimated river discharge from satellite-observed water elevation data by machine learning. Then, using the estimated discharge and water elevation, they performed the inversion of a hydrodynamic model to estimate bathymetry and other parameters by variational data assimilation.

General comments: Although the topic of this paper is suitable to NPG, I believe that this paper has some fatal flaws which cannot be fixed in the short period of time. I believe that the current version of the paper cannot be accepted.

First, the design of the authors' synthetic experiment is inappropriate. Their synthetic

observations were fully generated by hydrodynamic models with no observation and model errors. They may not consider the real satellite swath, and the temporal resolution of the data (daily) is much higher than the real satellite altimetry. I believe that they have too rich data to examine the potential of SWOT. The richness of the observation data significantly matters when the fully data-driven approach such as neural network is applied but the authors completely ignored this issue. I strongly recommend the authors to perform numerical experiments with more realistic data.

Second, the advantage of the proposed method is unclear for me. In my understanding, there are many methods to infer river discharge from water levels. The authors omitted to compare their neural network with those previous works so that I am not convinced that machine learning is necessary in this context. As the authors raised in section 1, there are many methods to perform river bathymetry by assimilating satellite altimetry observations into hydrodynamic models. In my understanding, some of them simply applied the flavors of Kalman filter and successfully inferred river bathymetry (and river discharge) using the real satellite data from ENVISAT, ICESAT, and JASON-2 (e.g., Breda et al. 2019 https://doi.org/10.1029/2018WR024010). The authors' method seems to be much more complicated than these previous works and I am not convinced that the complex processes are necessary. I strongly recommend the authors to perform many sensitivity analyses and to confirm the impact of each process on the performance of their method.

Specific comments: Major points: L113: section 2.1.3. should not be "In-situ data". The authors actually generated synthetic in-situ data by simulation. This is misleading.

L118: I believe that daily sampling data cannot be called "SWOT like" observations although it may be accepted in the previous papers.

L119: As mentioned above, the assumption of perfect observation is problematic.

L142-145: Why did you calculate Pearson correlation coefficient? The authors did not use this information in this paper.

L148-149: I could not understand why the authors excluded the data whose mean discharge is larger than 10000 m3. Since machine learning basically interpolates the data, it is generally recommended to make training data cover the wide range of state space. If they cannot have the access to those data, maybe they should not use fully data-driven approaches. Why should the authors choose the inappropriate experiment design?

L217, Table 2: I recommend the authors to use same metrics for Tables 1 and 2.

L440, Figure 10: How did the authors get the target of bathymetry (red dots)?

Minor points: L23: Maybe the authors can divide this paragraph around this line. This first paragraph is too long and includes several topics. L61: the estimations accuracy –> the estimation's accuracy L417: Please fix a typo ("Section ??").

---

## Short Comment (SC1) · 22 Sep 2020

This study aims at proposing the hybrid Neural Network (NN) – variational data assimilation algorithm to estimate river discharge from simulated SWOT like data. Such methodological studies are very important and of the scope of the NPG. In addition, investigating the potential benefits of satellites prior to the launches is quite useful to improve satellite missions further. However, I think the present manuscript has some fatal issues that should be solved prior to publication. The authors seemed to investigate the method that would not be applicable to the real ungauged river basins as I elaborate below. I am compelled to suggest this manuscript be rejected.

[Figure]

[Major Issues] 1. As described, the SWOT-based estimation of river discharge is useful for ungagged or poorly gauged river basins (P1L14). However, the authors used "too rich" basin information. They used dA (difference in cross-section), W (river width), S (slope), and A (cross-section) to estimate Q (discharge) by NN (P8L166). The physical-based models, which were also used to mimic observation data, simulate Q based on dA, W, S, and A with only one major uncertainty parameter: frictions of the river channel. Namely, there is one equation and one uncertain parameter. Solving this problem is too very easy for NN. Consequently, the present experimental setting of NN was very confusing to me. It is usually impossible to use the cross-section A because the cross-section under the river surface is unobservable by satellites. The challenge for realistic applications is to estimate Q without using A.

2. The authors assumed unrealistic daily SWOT observation data while real satellite revisits 1-4 times per 21 days (P1L22).

Consequently, I strongly suggest the authors re-consider experimental design that is applicable to real problems.

[Other Issues] 1. Experimental design is unclear to me. It is better to add a schematic image that shows the low chart of data used in this algorithm.

2. The paper should add more hydrological papers for reference. For example, I found a data-driven estimation of river width from satellite data (Yamazaki et al. 2014). Comparisons with such existing approaches would be beneficial to add the values of the manuscript. https://agupubs.onlinelibrary.wiley.com/doi/full/10.1002/2013wr014664

---

## Author Comment (AC1) · 25 Sep 2020

In this paper, the authors developed a new method for river bathymetry and dischargeestimation from satellite altimetry data. They firstly estimated river discharge fromsatellite-observed water elevation data by machine learning. Then, using the estimateddischarge and water elevation, they performed the inversion of a hydrodynamic modelto estimate bathymetry and other parameters by variational data assimilation.General comments: Although the topic of this paper is suitable to NPG, I believe thatthis paper has some fatal flaws which cannot be fixed in the short period of time.

Ibelieve that the current version of the paper cannot be accepted.

First, the design of the authors' synthetic experiment is inappropriate. Their synthetic observations were fully generated by hydrodynamic models with no observation and model errors. They may not consider the real satellite swath, and the temporal resolution of the data (daily) is much higher than the real satellite altimetry. I believe that they have too rich data to examine the potential of SWOT. The richness of the observation data significantly matters when the fully data-driven approach such as neural network is applied but the authors completely ignored this issue. I strongly recommend the authors to perform numerical experiments with more realistic data. **

> On the "inappropriate synthetic experiment". Data are synthetic, that is correct. These data have been generated by calibrated models (Hec-Ras and LisFlood in particular) by a relatively large community. They constitute the reference Pepsi and Pepsi-2 datasets of the SWOT Science Team. They contain $\sim$ 56 000 "examples" (in the machine learning sense) representing 29 heterogeneous rivers; this is a volume of reference data (considered reliable) never reached up to now in the community. These datasets are the most advanced ones in the SWOT community for assessing and benchmarking the different algorithms on a large panel of different rivers presenting different flow regimes, see [Durand et al. 2016, Frasson et al. 2020]. These data are not as realistic as those considered in [Tuozzolo et al. 2019] for example. Indeed, this last reference constitutes one of the first study (among very few today) based on real SWOT like data (airborne ones, NASA AirSWOT mission). (We, the authors of the present manuscript, have performed our previous version of algorithm for [Tuozzolo et al. 2019]; it was one of the two employed and compared algorithm). Considering real Airborne data or data from the SWOT science simulator (https://swot.jpl.nasa.gov/mission/overview/) does not affect the method-algorithms analysis. It might affect the accuracy of the estimations but not the crucial features of the method. Considering synthetic data with no noise, next with a given level Gaussian noise, is a mandatory step to better understand and evaluate a method capability. This

is what is done in the present study following the Discharge Algorithm Working Group research plan of the SWOT Science Team, see [Durand et al., 2016] and Frasson et al. 2020] Phase 1. Moreover some noisy data have been taken into account but the results were not shown in the present study (since the study focuses on Phase 1). However, following your remark we have added a few comments from the results we have obtained with noise. Please read our more detailed answer to your final comment on "sensitivity analysis".

On the "temporal resolution". That is correct, the considered data frequency is 1 day only. This corresponds to the important Cal-Val orbit phase of the satellite.

In short, the considered datasets are synthetic, 1-day repeat, covering a very large rivers sets with very different flow characteristics. This responds to an important science issue, at the forefront of the current Discharge Algorithm Working Group (https://swot.jpl.nasa.gov/documents/4050/)

You are right, these points were not sufficiently explained. Now, they are much better indicated throughout the manuscript, including in the new abstract, in the general introduction and conclusion, and of course in the data section too.

Note that if considering the nominal SWOT orbit (which will provide data with 21 days revisit period, depending on the latitude), the scientific challenge which consists to solve the ill-posed inverse problem for ungauged rivers posed by the mission remains the same (see Section 5.3 of the manuscript or our next answer to your comment). In this case, the time validity of the discharge estimation equals the wave travelling time through the river portion (roughly, a few hours to a day, depending on the case), see eg. [Tourian et al. 2017], [Brisset et al 2018], [Larnier et al. 2020] (with the identifiability map concept in particular). This point is well understood now. The present remark has been added in the dedicated new section 3.4 entitled "On the sensitivity of the estimations with respect to error measurements or time frequency".

On the "neural network" estimations. Recall that a standard ANN is interpolator but

based on a complex multi-resolution model (defined by its architecture) and a large volume of data. After optimization (training stage), the ANN has "identified-learned" invariants, correlations between the four input variables and the output variable Q, see eg. [Mallat (2016) Phil. Trans. R. Soc. A 374: 20150203]. As you know, the results obtained by ANN can be astonishing including in fluid mechanics, see eg. [Brenner, et al. (2019). Physical Review Fluids, 4(10), 100501], despite no one fully understand how it actually works yet.

In the present ANN, the concept of spatial correlation or time correlation between examples does not exist. Indeed, the ANN input variables are dA,W,S and Area; one "example" corresponds to a set of (4+1) values which are point-wise, snapshots. No space correlation nor time ones exist between two "examples". As a consequence, the ANN does not "see" potential space or time correlation between the datasets.

In our case, if considering less frequent observations (eg. with few days frequency), but of course with similar volume and quality of data, the accuracy of the trained ANN is similar. We have investigated this assertion for a frequency of 5 days (results not shown here). As expected the obtained accuracy were of same order of magnitude than those presented in Table 2. Obviously, in this case (eg. with 21 days revisit) and for the reason previously mentioned (identifiability map), the discharge estimations remain valid for a few hours - a day around the observation instant only. Note that we have performed many other tests demonstrating the robustness of the present ANN estimations and eg. their insensitivity to the test – train river sets.

A remark on these points (non-correlated feature of the examples and robustness of the ANN estimation for less frequent observations) has been added in the dedicated new section entitled "On the sensitivity of the estimations with respect to error measurements and time frequency".

Again, all the points you mention were not sufficiently clear in the manuscript, or even not mentioned for a few of them. All are now much better highlighted in the new version.

Please read the new abstract, the new general introduction, the new conclusion and the sections 2 and 3 in particular. **

Second, the advantage of the proposed method is unclear for me. In my understanding, there are many methods to infer river discharge from water levels.

> Correct; we have tried to present a relatively large bibliography in the general introduction. **

The authors omitted to compare their neural network with those previous works so that I am not convinced that machine learning is necessary in this context.

> We do not agree; the present study demonstrates that a machine learning approach (the ANN) can help to solve a crucial step in the inversions. Firstly, note that the mentioned bibliography in our introduction is quite complete; the unsolved issues are clearly explicited. Secondly, whatever the adopted physically-based inversions method (Kalman Filter or Variational Data Assimilation), one of the most remaining critical challenge is to determine theÂăprior information, in particular the first guess value(s) of these iterative algorithms, see eg. a related discussion in [Frasson et al 2020]. Until recently, this point was not really, or at least not sufficiently, discussed. (Note that this point becomes even more critical if the study relies on a single river only). In particular, the VDA physically-based approach enables to capture space-time variations like almost no other published method does (see the cited bibliography), however a shift remains: it is the bias we address in this article. Please re-read the abstract, the introduction and Section 5.3. in particular. After optimization, the bias value was depending strongly on the first guess value (and of the method of course too). Here, this crucial issue seems to be solved for ungauged rivers by a machine learning approach (the ANN) plus the hierarchical flow model for rivers belonging to the learning partition (denoted here by Q-Lset). This is new, robust and absolutely promising. This is evaluated and analyzed for a large number of rivers. Moreover, as clearly mentioned in the manuscript, the ANN estimation is not a "final product". The ANN estimation is greatly

improved first by the algebraic flow model, second by the VDA process (see this description eg. in the abstract). As a consequence, the estimation to be compared with the other approaches would be the final estimation and not this intermediate purely-data driven value. Moreover, this purely data-driven estimation is implicitly compared to the "final" estimation (since being the "rough" basic estimation"). And the latter is implicitly compared to others state-of-the-art methods through the previous articles eg. [Durand et al. 2016, Tuozzolo et al. 2019, Larnier et al. 2020, Frasson et al. 2020]. Recall that we have directly participated to likely the most extensive comparisons published up to now [Durand et al. 2016, Frasson et al. 2020], [Tuozzolo et al. 2019]; benchmarks based on a large number of synthetic rivers plus one of the very few Airborne dataset (AirSWOT mission). These comparisons have been performed with our former algorithm(s), the purely deterministic VDA method presented in [Brisset et al. 2018], [Larnier et al. 2020] (and implemented in the same computational code as the present one, a former version of course). As a consequence, this very solid experience of benchmarking has provided us numerous reference results to compare with. These experiences enable us to claim the results we obtain here (and to implicitly compare them to the mentioned studies). Obviously, the posed inverse problem is not fully solved; also, other benchmarks between different complimentary approaches will be organized soon. **

As the authors raised in section 1, there are many methods to perform river bathymetry by assimilating satellite altimetry observations into hydrodynamic models.

> Correct but only if one of the other unknown key parameter is provided. More precisely, if one has a good key prior information such as one discharge value or one reference bathymetry value (at a single location is enough), then it has already been demonstrated that the SWOT inverse problem can be solved with a reasonable accuracy, see the references cited in our general introduction. For ungauged rivers, the algorithms have to be able to infer the discharge and the bathymetry. As a consequence, the actual inverse problem is to infer the pair $(Q(x,t); b(x))$ plus of course a

corresponding effective friction parameter K (constant or not). Based on hydrodynamics models, this is an ill-posed inverse problem (see Section 5.3). This is the inverse problem addressed in the present manuscript, with, to our best knowledge, a capability to solve it never reached up to now. Note that the present study focuses on the quality of the discharge estimation. Another article in preparation focuses on the quality of the bathymetry estimation obtained by the algorithm. **

In my understanding, some of them simply applied the flavors of Kalman filter and successfully inferred river bathymetry (and river discharge) using the real satellite data from ENVISAT, ICESAT, and JASON-2 (e.g., Breda et al. 2019 https://doi.org/10.1029/2018WR024010).Âă

> Thank you for mentioning this very recent reference. However, the inverse problem addressed in [Breda et al.] is not the same as the present one: the authors infer the bathymetry only (plus an effective friction coefficient), with the discharge given. This inverse problem is mathematically much much easier. Moreover, this is not the encountered inverse problem in ungauged rivers cases. Indeed, see their supporting information TextS4, "the model was forced using in situ observations of discharge at GS 15400000 (upstream boundary condition) and water levels at GS 15940000 (downstream boundary condition)". The discharge value was imposed at upstream at daily frequency.

Actually, their inverse problem can be solved by others approaches-algorithms too; including the present algorithm, see [Larnier et al 2020]. (Note that this would be interesting to compare the available different methods to solve this inverse problem in this particular case). (Note that [Breda et al.] is original not for its classical Kalman Filter approach but for its global optimisation approach based on a genetic algorithm -the ÂńÂăSCE-UA algorithmÂăÂż-and the use of multi-satellite like data –which are synthetic too-). **

The authors' method seems to be much more complicated than these previous works

and I am not convinced that the complex processes are necessary.Âă

> The present method (partly) solves an inverse problem unsolved up to now (considering ungauged river without accurate prior information). The use of the final algorithm implemented into the open-source software DassFlow is not more complex than the use of standard computational hydro-informatics codes. The VDA approach may be qualified as complex in the hydrology community but it is a standard inversion approach in others geosciences community like oceanography for example (including in the SWOT community). Also, note that a dedicated toolchain based on HiVDI algorithm has been implemented (and validated) to automatically produced discharge estimations from standard datasets used in the SWOT community (datasets from the Pepsi challenges, or as those produced from AirSWOT data or from the SWOT Science Simulator). In other respect, as explained in Section 7, once a one year data assimilation has been processed (ie. after one year of the instrument acquisition), an extremely simple algebraic model can provide in CPU-real-time the discharge estimation. The critical scientific challenge is to learn ungauged observed rivers (without accurate prior information); this is a complex inverse problem; it seems to require sophisticated mathematical and numerical tools. At the end, our resulting operational system (the calibrated algebraic flow model, Section 7) is very simple. **

ÂăI strongly recommend the authors to perform many sensitivity analyses and to confirm the impact of each process on the performance of their method

> Sensitivity analyses have been performed at each stage of the study and for each stage of the global algorithm: ANN, the algebraic flow model, VDA-based on the St-Venant equations. Sensitivity analyses (equivalently, inversions robustness) have been previously addressed both for the algebraic flow model and the VDA approach, see [Brisset et al. 2018, Tuozzolo et al. 2019, Larnier et al. 2020, Frasson et al. 2020].

Sensitivity analyses on the ANN step only had been performed but was not shown. As expected, the resulting accuracy is slightly degraded but the robustness remains.

Following your comment, a remark on this point (estimations with Gaussian noise with the expected instrument accuracy) have been added in the new subsection 3.4 entitled "On the sensitivity of the estimations with respect to error measurements and time frequency". A thorough analysis of the complete inversion algorithm sensitivity in a context of real like (provided by a simulator or from the AirSWOT campaing aforementioned) should be done during the next benchmarking study. * Our past experiments (including those based on the aforementioned real datasets) plus the present ones have convinced us that the presented scientific approach is solid and partly answers to an open problem unsolved up to now.

Following your remarks, we added in the new version numerous clarifications. This should make the manuscript clearer, in particular by better highlighting the context and the academic feature of the numerical experiments. Recall however that he latter are a mandatory step; moreover they clearly show a very solid improvement of the discharge estimation in this SWOT like context.

We sincerely thank you for your comments which have greatly help to clarify the hydrology problem, the approach capabilities and the limitations of the study, therefore the necessary forthcoming studies to assess estimations for ungauged rivers from eg. the SWOT simulator. **

Specific comments:Âă Major points:

L113: section 2.1.3. should not be "In-situ data". The authors actually generated synthetic in-situ data by simulation. This is misleading.

> Data are synthetic, that is correct. ÂăPlease, see the previous discussion for details. To be more accurate, we have replaced the term "in-situ data" by "in-situ type data" throughout the text (including in the subsection name of course). Moreover, as already mentioned above, data origins and features have been recalled more clearly (including in the abstract). **

L118: I believe that daily sampling data cannot be called "SWOT like" observations although it may be accepted in the previous papers.

> This point has been better highlighted throughout the text, including in the abstract. We explicitly refer now to the Cal-Val phase of the instrument and to Phase 1 of the so-called "Pepsi challenge" defined in [Durand et al. 2016, Frasson et al., 2020]. Please, see the previous discussion. **

L119: As mentioned above, the assumption of perfect observation is problematic. > Please, see the previous related discussion too. **

L142-145: Why did you calculate Pearson correlation coefficient? The authors did not use this information in this paper.Âǎ Âǎ >We compute the R2 correlation criteria because it is a good (and classical) performance criteria to measure the efficiency of an ANN prediction vs the true values. This feature fully applies to the experiments presented in Tab. 1. **

L148-149: I could not understand why the authors excluded the data whose mean discharge is larger than 10 000 m3. Since machine learning basically interpolates the data, it is generally recommended to make training data cover the wide range of state space. If they cannot have the access to those data, maybe they should not use fully data-driven approaches. Why should the authors choose the inappropriate experiment design?

> This is not an ÂńÂǎinappropriate experimentÂǎÂż. You are right, as previously mentioned a well trained ANN can accurately represent multi-scale, highly non-linear observed phenomena, in a least-square sense. You are right, a trained ANN constitutes an excellent interpolator but a-priori not an extrapolator (ie. out of the learning range values). Here, one expect that its prediction capabilities hold within the learning partition Q-Lset only. In our case, the preliminary statistical analysis show that the great majority of "examples" (in the sense of Machine Learning i.e. dataset at one location) presents a mean discharge value lower than 10 000m3/s. The few rivers presenting

mean discharge values greater than 10 000m3 are somehow outliers; they represent less than 10% of the examples. Then we have designed the experiments to show: 1) the capability of an ANN to (roughly) estimate a discharge from the (3+1) input variables only. 2) the (un-)capability of estimations for larger rivers for which one could not acquire data enough to train the ANN. This is our experiment plan and goals. The obtained conclusions seem clear and robust.

Basically, given a (unmonitored) river presenting the same characteristics as the learning partition, one can expect a more accurate ANN estimation. (Here, the partition Q-Lset contains the rivers with mean discharge lower than 10 000 m3/s). This intuitive feature is verified here, in the present case. We have tested (results not shown here): one obtain a better ANN model (or at worse a similar) for rivers belonging to the training partition, than if training for the whole values range.

However, if one train the ANN from the complete range of discharge values like you suggest it (ie. without excluding discharge values greater than 10 000, namely Jamina, Mississipi downstream, Padma), then the ANN predictions for these three rivers are much more accurate. But, recall, that in that case, the same ANN is less accurate or (at best similar) for the other rivers ie. those belonging to Q-Vset-in. This is what show (confirm) the present numerical experiments.

In practice and if one approximatively knowns at what class a river belongs to (this assumption is reasonable for a great majority of rivers in the world eg. from the GRDC database), and if one have data enough to perform a good training process, then it seems to be more efficient to train the ANN with "examples" (datasets) from the corresponding class of rivers. This is what we suggest.

Note that detailing all these numerical experiments and the resulting properties cannot be done in the same article. We believe that the present manuscript demonstrates already a lot of new estimations capabilities and intrinsic properties of the different elaborated algorithms (the ANN, the algebraic flow model and the advanced VDA pro-
cess).

However, again following your remark, the experiment plan and its goal are now much more detailed, see the new dedicated section 2.4 entitled "On the choice to define two river classes". **

L217, Table 2: I recommend the authors to use same metrics for Tables 1 and 2.

Âň As already mentioned, the Pierson correlation coefficient (R2) fully makes sense for the Table 1 experiment since based on the complete set. It is more questionable for small datasets like it is the case for a single river only. That is why it has not been indicated neither in Table 2 nor in Table 3. On the contrary computing the nRMSE or the NSE for the whole dataset is meaningless. These criteria make sense for each river. For Table 1 experiment, the NSE represents a mean value only (like the nRMSE), that is why we initially choose to not indicate it. However, following your remark this criterion is indicated in Table 1 now.

L440, Figure 10: How did the authors get the target of bathymetry (red dots)?

> The target bathymetry values are those employed in the various calibrated reference flow models (HEC-Ras, LisFlood etc) which have been performed to obtain the synthetic data available in the Pepsi 1 and Pepsi 2 datasets, see [Durand et al. 2016, Frasson et al. 2020] and references therein. Here, the red dots are computed from the effective rectangular values of the unobserved lowest cross-section A0 (W=W0, H0=Z0-b). As indicated in Section 4.3.1, these "true" (= reference model values) are available at the Reference Data Scale only. This point is better detailed now, see Section 4.3.1 and the end of the introduction of Section 6. **

Minor points:Âă L23: Maybe the authors can divide this paragraph around this line. This first paragraph is too long and includes several topics.Âă > corrected. L61: the estimations accuracy –> the estimation's accuracyÂă > corrected. L417: Please fix a typo ("Section ??"). > corrected.

Thank you for your detailed proof reading.

(The aforementioned new version of manuscript should be posted online within 2-3 working days).
* * *

---

## Author Comment (AC2) · 25 Sep 2020

This study aims at proposing the hybrid Neural Network (NN) – variation-al data assimilation algorithm to estimate river discharge from simulated SWOT like data. Such methodological studies are very important and of the scope of the NPG. In addition, investigating the potential benefits of satellites prior to the launches is quite useful to improve satellite missions further. However, I think the present manuscript has some fatal issues that should be solved prior to publication. The authors seemed to investigate the method that would not be applicable to the real un-gauged river basins as elaborate below. I am compelled to suggest this manuscript be rejected.

[Figure]

[Major Issues] 1. As described, the SWOT-based estimation of river discharge is useful for ungagged or poorly gauged river basins (P1L14). However, the authors used "too rich" basin information. They used dA (difference in cross-section), W (river width), S(slope), and A (cross-section) to estimate Q (discharge) by NN (P8L166).

> Undoubtedly, there is a misunderstanding; moreover there were an error of typing p8l66 ! First, the letter \cal A does not denote the wetted area (hopefully. . .); it denotes the (local) drainage area. The wetted cross-section area is denoted by A. The ANN input variables are (dA,W,S) and \cal A. This was indicated in the abstract, in the introduction (p2l58), p6l135, etc. This was not recalled p8l66; now it is. Moreover, p8l66, an unfortunate copy-paste was present: obviously, the knowledge of dA does not imply the knowledge of A0 ! **

The physical-based models, which were also used to mimic observation data, simulate Q based on dA, W, S, and A with only one major uncertainty parameter: frictions of the river channel. Namely, there is one equation and one uncertain parameter. Solving this problem is too very easy for NN. Consequently, the present experimental setting of NN was very confusing to me. It is usually impossible to use the cross-section A because the cross-section under the river surface is unobservable by satellites. The challenge for realistic applications is to estimate Q without using A.2.

> We agree; if one had considered the observations of A0 (or equivalently of the complete wetted cross-section A), the inverse problem would be much much easier to solve ! Obviously, this is not the case (see above). The considered inverse problem is the most complete and the most challenging one, in the present altimetry context. This inverse problem is those indicated throughout the manuscript eg. in the abstract, in the general introduction and eg. in Section 5. **

The authors assumed unrealistic daily SWOT observation data while real satellite revisits 1-4 times per 21 days (P1L22). Consequently, I strongly suggest the authors re-consider experimental design that is applicable to real problems.

> That is correct, the considered data frequency is 1 day only. This corresponds to the important Cal-Val orbit phase of the satellite.

In short, the considered datasets are synthetic, 1-day repeat, covering a very large rivers sets with very different flow characteristics. This responds to an important science issue, at the forefront of the current Discharge Algorithm Working Group (https://swot.jpl.nasa.gov/documents/4050/)

These points were not sufficiently explained. Now, they are much better indicated throughout the manuscript, including in the new abstract, in the general introduction and conclusion, and of course in the data section too.

Note that if considering the nominal SWOT orbit (which will provide data with 21 days revisit period, depending on the latitude), the scientific challenge which consists to solve the ill-posed inverse problem for ungaged rivers posed by the mission remains the same (see Section 5.3 of the manuscript or our next answer to your comment). In this case, the time validity of the discharge estimation equals the wave travelling time through the river portion (roughly, a few hours to a day, depending on the case), see eg. [Tourian et al. 2017], [Brisset et al 2018], [Larnier et al. 2020] (with the identifiability map concept in particular). This point is well understood now. The present remark has been added in the dedicated new section 3.4 entitled "On the sensitivity of the estimations with respect to error measurements or time frequency". **

[Other Issues] 1. Experimental design is unclear to me. It is better to add a schematic image that shows the low chart of data used in this algorithm.

> Thank you for this suggestion. We have added in the introduction of Section 5, a figure representing synthetically the complete algorithm (ANN, low complexity alge-braic flow model, VDA process based on the St-Venant equations and finally real-time estimations from newly acquired data), the employed data and the priors of the com-putational method. **

2. The paper should add more hydrological papers for reference. For example, I found a data-driven estimation of river width from satellite data (Yamazaki et al. 2014). Comparisons with such existing approaches would be beneficial to add the values of the manuscript. https://agupubs.onlinelibrary.wiley.com/doi/full/10.1002/2013wr014664

> Thank you for mentioning us this reference which aims at "calculating river width from satellite‐based water masks and flow direction maps". This is not the topic of the present study. This study could be an alternative source of the data variable W. For this reason, we mention it now in the data section.

(The aforementioned new version of the manuscript should be posted online within 2-3 working days).

---

## Author Comment (AC4) · 3 Oct 2020

[12pt]article

*This study aims at proposing the hybrid Neural Network (NN) – variation-al data assimilation algorithm to estimate river discharge from simulated SWOT like data. Such methodological studies are very important and of the scope of the NPG. In addition, investigating the potential benefits of satellites prior to the launches is quite useful to improve satellite missions further. However, I think the present manuscript has some fatal issues that should be solved prior to publication. The authors seemed to investigate the method that would not be applicable to the real un-gauged river basins as elaborate below. I am compelled to suggest this manuscript be rejected.*

[Figure]

*[Major Issues] 1. As described, the SWOT-based estimation of river discharge is useful for ungagged or poorly gauged river basins (P1L14). However, the authors used "too rich" basin information. They used dA (difference in cross-section), W (river width), S(slope), and A (cross-section) to estimate Q (discharge) by NN (P8L166).*

\*

Undoubtedly, there is a misunderstanding; moreover there was a typing error p8l66. The letter $\rfloor\dashv\updownarrow\mathcal{A}$ (= cal A in the manuscript) does not denote the wetted area (hopefully...); it denotes the (local) drainage area. The wetted cross-section area is denoted by $A$.
The ANN input variables are (dA,W,S) and $\rfloor\dashv\updownarrow\mathcal{A}$ (= cal A in the manuscript).
All this was indicated in the abstract, in the introduction (p2l58), p6l135, etc.
This was not recalled p8l66; now it is. Moreover, p8l66: obviously, the knowledge of $dA$ does *not* imply the knowledge of $A_0$ (typing error !?...).

\*

*The physical-based models, which were also used to mimic observation data, simulate $Q$ based on $dA$, $W$, $S$, and $A$ with only one major uncertainty parameter: frictions of the river channel. Namely, there is one equation and one uncertain parameter. Solving this problem is too very easy for NN. Consequently, the present experimental setting of NN was very confusing to me. It is usually impossible to use the cross-section A because the cross-section under the river surface is unobservable by satellites. The challenge for realistic applications is to estimate Q without using A.2.*

We agree; if one had considered the observations of $A_0$ (or equivalently of the complete wetted cross-section A), the inverse problem would be much much easier to

solve ! Obviously, this is not the case (see above).

The considered inverse problem is the most complete and the most challenging one, in the present altimetry context. This inverse problem is those indicated throughout the manuscript eg. in the abstract, in the general introduction and eg. in Section 5.

*

*The authors assumed unrealistic daily SWOT observation data while real satellite revisits 1-4 times per 21 days (P1L22). Consequently, I strongly suggest the authors re-consider experimental design that is applicable to real problems.*

That is correct, the considered data frequency is 1 day only. This corresponds to the important Cal-Val orbit phase of the satellite (also called the "science orbit").

In short, the considered datasets are synthetic, 1-day repeat, covering a very large rivers sets with very different flow characteristics. This responds to an important science issue, at the forefront of the current Discharge Algorithm Working Group (https://swot.jpl.nasa.gov/documents/4050/).

These characteristics were apparently not sufficiently highlighted. Following your comment, they are now much better indicated throughout the manuscript, including in the new abstract, in the general introduction and conclusion, and of course in the data section too.

Note that if considering the nominal SWOT orbit (which will provide data with 21 days revisit period, depending on the latitude), the scientific challenge which consists to solve the ill-posed inverse problem for ungauged rivers posed by the mission remains the same (see Section 5.3 of the manuscript or our next answer to your comment).

In this case, the time validity of the discharge estimation equals the wave travelling time through the river portion (roughly, a few hours to a day, depending on the case), see eg. [Tourian et al. 2017], [Brisset et al 2018], [Larnier et al. 2020] (with the identifiability map concept in particular). This point is well understood now.

The present remark has been added in the dedicated new section 3.4 entitled "On the sensitivity of the estimations with respect to error measurements or time frequency".

*

*[Other Issues] 1. Experimental design is unclear to me. It is better to add a schematic image that shows the low chart of data used in this algorithm.*

Thank you for this suggestion. We have added in the introduction of Section 5, a figure representing synthetically the complete algorithm (ANN, low complexity algebraic flow model, VDA process based on the St-Venant equations and finally real-time estimations from newly acquired data), the employed data and the prior of the computational method.

*

*2. The paper should add more hydrological papers for reference. For example, I found a data-driven estimation of river width from satellite data (Yamazaki et al. 2014). Comparisons with such existing approaches would be beneficial to add the values of the manuscript. https://agupubs.onlinelibrary.wiley.com/doi/full/10.1002/2013wr014664*

Thank you for pointing us this reference which aims at "calculating river width from satellite based water masks and flow direction maps". This is not the topic of the present study. This study could be an alternative source of the data variable W. For this reason, we mention it now in the data section. Moreover we have since detected a
few more connected articles from the hydrology community (on the use of ANN and a new useful database); we have added them.

---

## Author Comment (AC5) · 3 Oct 2020

**Hybrid Neural Network - Variational Data Assimilation algorithm to infer river discharges from SWOT-like data**

Kevin LARNIER[1][2][3] and Jérôme MONNIER[1][2]

[1]Institut de Mathématiques de Toulouse (IMT), France
[2]INSA Toulouse, France
[3]CS corporation, Business Unit Espace, Toulouse, France

**Correspondence:** J. Monnier (jerome.monnier@insa-toulouse.fr)

**Abstract.** A new algorithm to estimate discharges from altimetry measurements is designed for rivers with unknown bathymetry and no prior flow information. An issue in data assimilation approaches is to define a good first guess. Indeed, it is shown that if computing from the classical hydro flow models, the inverse problem may be well-defined but up to a bias (the bias scales the global estimation). This key issue is tackled by performing an artificial neural network trained on altimetric large scale

5  water surface measurements plus a local drainage area information. The combination of this purely data-driven estimation with a dedicated algebraic flow model provides a first good physically-consistent estimation. The latter is next employed as the first guess of an advanced Variational Data Assimilation (VDA) formulation which enables to accurately capture space-time variations of the flow. For rivers belonging to the machine learning partition, the first guess value is accurate, therefore the bias of the final VDA estimation vanishes. For rivers outside the learning partition, the first guess is naturally less accurate; a bias

10  remains. However, in this case, any mean value (eg. annual or seasonal) can be assimilated to remove it. Numerical experiments are performed for 29 heterogeneous river portions. The water surface measurements are synthetic with a 1-day repeat, corresponding to the calibration-validation orbit phase of the forthcoming SWOT mission. The estimations obtained by performing this new hybrid hierarchical inversion method are robust. They are 
[revised manuscript text omitted]

During the Calibration-Validation (Cal-Val) phase (also referred as the "science orbit"), the instrument will have a 1-day repeat orbit.

In the present study, the considered data are SWOT-like ones during this Cal-Val phase, Rodriguez and others (2012); Rodriguez and Esteban-Fernandez (2010), therefore data of the same nature as the forthcoming nominal SWOT data but with a 1-day revisit. Moreover as a first step and following the Pepsi 1 and Pepsi 2 Durand et al. (2016); Frasson et al. (Submitted) benchmarks design (Discharge Algorithm Working Group of the SWOT Science Team), we consider perfect data. The next phase will consist to introduce realistic noises; the last phase consists to consider outputs from the SWOT Science simulator or data from AirSWOT (AirBorne) campaigns. This scientific approach enables to rigorously analyse the methods capabilities (or not) to tackle the present ill-posed inverse problem.

120  Future works will be dedicated to real SWOT-like data with realistic errors and temporal samplings; for instance, data computed using the SWOT Science Instrument simulator or acquired during a AirSWOT (airborne) campaigns. Note such study using the former version of HiVDI algorithm (VDA based only) has been conducted in Tuozzolo et al. (2019).

Here the considered data are as follows:

125  – The complete set of measurements $(Z_{r,p}, W_{r,p}, S_{r,p})$ at SwReachSc for each reach $r$ and at each instant $p$.

– The measurements of $(Z_{r,p}, W_{r,p})$ at RefDataSc for each "node" $r$ and at each instant $p$.

In the sequel and if ambiguous, it will be clarified at which scale the different fields and data are considered.

The SWOT instrument should provide WS measurements $(Z, W)$ at the "node scale" $200m$ long. This fine scale data is represented by data available in the Pepsi 1 and Pepsi 2 databases at RefDataSc.

130  Each river portion is decomposed into $R$ reaches: $r = 1, .., R$, Fig. A1. It is assumed that $(P + 1)$ instants of measurements are available; the corresponding measurements are ordered by flow elevations $Z$; the case $p = 0$ denotes the lowest water level and $p = (P + 1)$ denotes the highest.

Given a river portion, the resulting SWOT data set is $\{Z_{r,p}, W_{r,p}\}_{R,P+1}$ plus WS slope $\{S_{r,p}\}_{R,P+1}$ at SwReachSc.

Depending on the considered flow model, the $r$-th "spatial point" denotes either the node or the reach number. More pre-
135  cisely, the node scale is the adequate scale for the Saint-Venant flow model B1, while the larger reach scale is consistent with the low complexity algebraic model 5, see Garambois and Monnier (2015); Brisset et al. (2018) for investigations.

Note that SW elevations $Z$ may be obtained from multiple altimetry missions databases, see eg. DAHITI database Schwatke et al. (2015), however at heterogeneous accuracy and frequencies; also, rivers width $W$ may be extracted from eg. the Global
140  With Database built in Yamazaki et al. (2014).

Finally, let us point out that if considering the nominal SWOT orbit (21 days repeat orbit) with realistic noise, the scientific challenge which consists to solve the considered inverse problem remains of same nature as the present one; of course with higher uncertainties and a time limitation of the discharge estimations (see details at the end of next section).

145

**2.1.3  Synthetic in-situ data**

**2.1.4  References data: Pepsi datasets**

To generate synthetic data of discharges, we employ the Pepsi databases which have been built up for the Pepsi 1 and 2 challenges, see Durand et al. (2016); Frasson et al. (Submitted). These databases are a compilation of synthetic flow observations
150  generated from outputs of various hydraulic flow models. These models have been calibrated. It is assumed by the Discharge Algorithm Working Group of the SWOT ScienceTeam that these models represent sufficiently well the flow to constitute

references for benchmarking discharge algorithms, Durand et al. (2016); Frasson et al. (Submitted). The present SWOT like observations have been generated from these flow models outputs at daily sampling (corresponding to the CalVal orbit phase), both at RefDataSc and at SwReachSc (see previous paragraph). For the aforementioned reasons, here no errors have been added to the models outputs.

They contain numerous river portions with various hydro-geomorphological properties and various regimes. Other more sophisticated datasets exist (SWOT Instrument Simulator, AirSWOT campaigns data, see e.g. Tuozzolo et al. (2019), but they are restricted to very few river portions only. As a consequence, such datasets are not well suited for machine learning experiments.

The number of days, nodes and reaches varies from one river portion to another. The number of days varies from 12 days to a full year. The number of nodes by river portion varies from 21 to 3189; the number of reaches varies from 4 to 16. Some of the river portions in this dataset were outside the range of SWOT visibility since the width was less than 50m; they were then removed from the dataset. Similarly river portions with less than 100 days of observations were removed. Finally, a total number of 29 river portions were selected which represents a total count of 145 reaches and (time multiplied by space) of 55 525 observations of any variable at SwReachSc. At RefDataSc, values of $(Z, W)$ and $(Q, A)$ are available. At SwReachSc, values of $(Z, W, S)$ and $(Q, A)$ are available.

**2.1.5 Ancillary data**

To train the ANN, we will use local drainage area values (denoted by $\mathcal{A}$) as an input variable. In the present altimetry context, this variable may be considered as an ancillary data. The knowledge of $\mathcal{A}$ will be the only prior of the forthcoming inversion algorithm.

[revised manuscript text omitted]

**2.4    On the choice to define two river classes: $Q$-Lset and its complement**

The preliminary statistical analysis show that the great majority of "examples" (in the sense of machine learning) presents a mean discharge value lower than 10000 $m^3/s$. The rivers presenting mean discharge values greater than 10000 $m^3/s$ represent 210 less than 10% of the examples; here, they are kind of outliers. Then, for two reasons, we have divided the river sets in two distinguished classes: $Q$-Lset and its complement $Q$-Vset-out.

Firstly, because one expect to obtain a better ANN model if its training class is the same as those of the river test eg. an ANN trained on $Q$-Lset employed as a predictor for a river belonging to $Q$-Vset-in. Secondly, to evaluate the extrapolation capabilities (or not) of the ANN for rivers outside of the training class i.e. $Q$-Vset-out.

215 The numerical results presented in next section will illustrate the capabilities of the ANN for these different cases.

Note that classifying rivers within rough classes in terms of mean discharge value is realistic, for instance from the GRDC database or the GRADES database, see Lin et al. (2019).

**3    Data-driven estimations of $Q$ by ANN**

220 In this section, purely data-driven estimations of discharge are performed and analyzed. The estimations are obtained by training an Artificial Neural Networks (ANN) on the learning set $Q$-Lset.

**3.1    The ANN description**

The employed ANN is designed as follows. The training dataset $\mathcal{D}$ contains $N_{lp}$ learning pairs ("examples") $(I_i, Q_i)$, $i = 1, \cdots, N_{lp}$ . The $i$-th input is $I_i = (dA, W, S, \mathcal{A})_i$ where $i$ denotes the $i$-th value at the considered location and day. Recall that 225 $dA$ $(m^2)$ denotes the variations of the wetted crossed-sections above the unobserved value $A_0$; it is straightforwardly computed from the variations of $Z$ $(m)$ and $W$ $(m)$. The slopes values $S$ are extracted from the Pepsi databases (model outputs) at SwReachSc. Values of $\mathcal{A}$ $(km^2)$ are extracted from HydroSHEDS database.

Measurements are daily sampled. This frequency corresponds to the important Cal-Val phase of the forthcoming SWOT instrument. However, note that the data could be less frequent (provided eg. every few days) without altering the ANN accuracy; this point is discussed at the end of the section.

[revised manuscript text omitted]

290 for rivers outside the learning partition. In other words, the ANN makes an excellent (and often an amazing) interpolator but as expected not an accurate extrapolator. However, these first ANN estimations will be employed in the sequel as a prior.

Finally let us recall that one should generally be able to classify rivers by rough classes in terms of mean discharge from eg. the GRADES database, Lin et al. (2019). As a consequence, for a majority of rivers portions, one should be able to perform the ANN estimation within the learning partition and not outside, like it is done here.

[Figure]

**Figure 6.** Discharge values estimated by the trained ANN for river portions belonging to $Q$-Vset-out that is rivers presenting discharge values outside the learning partition. (Top Left) Jamuna. (Top Right) Mississippi downstream. (Bottom) Padma.

**3.4 On the sensitivity of the estimations with respect to error measurements and data frequency**

As already mentioned, as a first step and following the Pepsi 1 and Pepsi 2 Durand et al. (2016); Frasson et al. (Submitted) benchmarks design (from the Discharge Algorithm Working Group of the SWOT Science Team), algorithms evaluations are performed on data with no noise and 1 day sampling. Recall that the first SWOT observations will have a 1 day repeat period during the Cal/Val phase. The next phase of algorithms evaluations consists to introduce realistic noise and time sampling. Finally the last phase consists to consider outputs from the SWOT Science Simulator or data from AirSWOT (AirBorne) campaigns. This gradual approach is scientifically necessary to demonstrate the estimations capabilities of the methods, their robustness and their weakness or inaccuracy origins. This is the approach we rigorously follow here to evaluate the present new method capabilities.

However, we have performed numerous tests to be confident in the method robustness, its relative insensitivity with respect to a few data properties. First, we have performed numerous ANN estimations with different test and train sets; demonstrating the robustness (and its relative accuracy) of the purely data-driven estimations. Second, we have tested the sensitivity of the present ANN estimations with respect to : a) the noise on the WS measurements $(Z, W)$ ; b) the data frequency. The results

analysis are presented below.

*With noisy WS measurements $(Z, W)$.* We have tested the ANN estimation if considering perturbed measurements $(Z, W)$, respecting the expected instrument accuracy. Data have been perturbed by Gaussian noises with $\sigma_Z = 0.25$m (resp. $\sigma_W = 5.0$m) for $Z$ (resp. $W$) for 1-day repeat. The obtained results are as follows: NRMSE equals $18.1\%$ and NSE equals $0.91$. These results have to be compared with those indicated in Table 1, that is: $18.1\%$ vs $12.8\%$ and $0.91$ vs $0.98$. These results show that the ANN estimations remain robust to the inaccuracy of the WS measurements.

*With less frequent WS measurements.* In the present ANN, the concept of spatial correlation or time correlation between the examples does not exist. Indeed, each "example" corresponds to a set of $(4 + 1)$ values which are correlated neither in time nor in space; they are simple point-wise snapshots. As a consequence, the ANN does not "see" any space or time structure in the datasets. After optimization (training), an optimal ANN model has been built. The latter enables to reproduce invariants between the four input variables and the output variable Q. This is a classical amazing feature of ANNs, see eg. Mallat (2016), despite no one fully understand how it actually works yet. The ANN validation consists to test on numerous aleatory train - test datasets. This is what it has classically been done here.

Following the argument above, the present ANN has to provide a similar accuracy if considering much less frequent observations, but with the same volume and same quality of data of course. We have investigated this assertion for a frequency of 5 days. As expected the obtained accuracy were similar to those presented in Table 2. Note that the performances of ANN experiments are never exactly the same since based on aleatory features.

Obviously, in this case, the discharge estimations remain valid for a few hours - a day around the observation instant only. Indeed let us recall that the time validity of the discharge estimation is approximatively equal to the wave travelling time through the river portion (roughly, a few hours to a day), see eg. Paiva et al. (2015); Brisset et al. (2018); Larnier et al. (2020a) for such a discussion.

Let us recall that if considering the nominal SWOT orbit (21 days repeat orbit) with noisy data, the scientific challenge which consists to solve the inverse problem for ungaged rivers remains of same nature as the present one; of course with higher uncertainties and a time limitation of the discharge estimations (which equal the wave travelling time through the river portion).

**4 Physically-based estimations using the algebraic flow model: first guesses**

In this section, the low Froude flow model is presented; it is an algebraic system. The Strickler friction coefficient $K$ has to depend on space and time, then to reduce its complexity, it is modeled as a power-law in water depth $h$. Next given the WS

measurements and $Q^{(ANN)}$, the algebraic flow model is solved to obtain estimations of $(A_{0,r}, (\alpha, \beta)_r)$ and $Q_{in,p}$ $\forall r, p$. These estimations will be next considered as the first guess values in the VDA based inversion presented in next section.

**4.1 Reduced parametrization of $K$**

Following Garambois et al. (2020); Larnier et al. (2020a), the Strickler friction coefficient $K$ is defined as local power-laws at

345 SwReachSc: $K_{r,p} \equiv K((\alpha_r, \beta_r); h_{r,p}) = \alpha_r (h_{r,p})^{\beta_r}$ $\forall r \forall p$. As a consequence, given $R \times (P+1)$ measurements $Z_{r,p}$, the friction parameter $K_{r,p}$ is represented by $2R$ parameters only: $(\alpha_r, \beta_r)_{1 \leq r \leq R}$. This reduced parametrization provides a local effective power-law in $h$. The law reads in function of the WS measurements as:

$$K_{r,p} \equiv K((\alpha_r, \beta_r); A_{0,r}, W_{r,0}, Z_{r,p}) = \alpha_r \left( Z_{r,p} - Z_{r,0} + \frac{1}{W_{r,0}} A_{0,r} \right)^{\beta_r} \quad \forall r \forall p \tag{4}$$

350

In the sequel if one refers to the friction parameter $K_{r,p}$, this actually refers to its parametrization defined by (4).

**4.2 The algebraic flow model**

While deriving the flow equations (mass and momentum conservation laws), the Low Froude assumption ($Fr^2 << 1$) is applied. The resulting model is an algebraic system of $R$ equations (one equation per reach $r$); each equation is similar to the

355 Manning-Strickler law, see Larnier et al. (2020a); Brisset et al. (2018). Since this "Low Froude" flow model is algebraic, its complexity is low. Using the present reduced parametrization (4), this system reads as follows:

$$Q_{r,p}^{\frac{3}{5}} = \alpha_r^{3/5} (c_{r,p}^{(1)} A_{0,r} + c_{r,p}^{(2)}) \left( c_r^{(4)} A_{0,r} + c_{r,p}^{(3)} \right)^{3/5\beta_r} \quad 1 \leq r \leq R, \quad 0 \leq p \leq P \tag{5}$$

The coefficients $c_{r,p}^{(k)}$, $k = 1, \cdots, 3$, and $c_r^{(4)}$ can be evaluated from the altimetry measurements. Their expressions are:

$$c_{r,p}^{(1)} = W_{r,p}^{\frac{-2}{5}} S_{r,p}^{3/10}, \quad c_{r,p}^{(2)} = c_{r,p}^{(1)} \delta A_{r,p}, \quad c_{r,p}^{(3)} = (Z_{r,p} - Z_{r,0}), \quad c_r^{(4)} = \frac{1}{W_{r,0}} \tag{6}$$

360 System (5) constitutes the so-called algebraic flow model. It contains $R(P+1)$ equations.

If considering the full set of unknowns $((\alpha_r, \beta_r), A_{0,r}, Q_{r,p})$ i.e. $R(3 + (P+1))$ unknowns, it is an underdetermined system therefore admitting an infinity of solutions.

If the discharge values $Q_{r,p}$ are given, the system admits an unique solution for the two other variables $((\alpha_r, \beta_r), A_{0,r})$ ($2R$ unknowns). This is the way the first guesses $(K_{r,p}, A_{0,r})^{(0)}$ are computed given $Q_{r,p}^{ANN}$, see Section 4.3.

365 Moreover this system will be employed differently to compute real-time estimations of $Q$, see Section 7.

Finally it is worth to notice that if $A_{0,r}$ is given $\forall r$ (therefore all wetted areas $A_{r,p} = A_{r,0} + \delta A_{r,p}$ $\forall r \forall p$ are given) then by solving the algebraic flow model (5) the inference of the *ratio* $(Q/K)_{r,p}$ is possible but not the sough variables $(Q_{r,p}, K_{r,p})$. (Of course, this remark applies to the classical scalar Manning-Strickler's law too).

[Figure]

**Figure 7.** Effective Strickler friction coefficient $K$ computed by solving the low Froude (algebraic) flow model (5), given data of the considered river portions.

**Effective low Froude flow Strickler values**

370    Given the datasets presented in Section 2.2, the friction coefficient $K$ corresponding to the low Froude flow model is computed by solving (5), see Fig. 7. This is an effective low Froude Strickler coefficient. This plot highlights the large range value of the effective low Froude Strickler coefficient; also it confirms physically-consistent values of $K$ obtained from the other measurements.

**4.3    First guesses $(A_{0,r}, (\alpha, \beta)_r)^{(0)}$ and $Q_{in,p}^{(0)}$**

375    In next section the VDA formulation is presented. It aims at estimating the unknown "input parameters" of the Saint-Venant flow model which are: the time-dependent discharge at inflow $Q_{in}(t)$, the bathymetry $b(x)$ (equivalent to $A_0(x)$) and the friction coefficient $K$ (parametrized as $K(h(x,t)$, see (4)). The VDA algorithm is iterative; the choice of a good first guess is important. Below is presented how the first guess values $(A_{0,r}, (\alpha, \beta)_r)^{(0)}$ and $Q_{in,p}^{(0)}$ are computed.

**4.3.1    First guess $(A_{0,r}, (\alpha, \beta)_r)^{(0)}$**

380    Given the WS measurements and $Q^{(ANN)}$ (the discharge estimation obtained by ANN, see (3)), values of $(A_0(x), K)$ are estimated by solving the algebraic flow model (5). These values provide the first guesses values $(A_0^{(0)}(x), K^{(0)}(h(x,t)))$ in the VDA algorithm. Recall that the Strickler friction coefficient $K$ is space-time dependent through the reduced parametrization (4). Given $Q_{r,p}^{ANN}$ (discharge estimation for reach $r$ at instant $p$), the algebraic system is solved by using the Metropolis-Hasting algorithm (MCMC method) to obtain $(A_{0,r}, (\alpha_r, \beta_r))^{(0)}$.

In the Metropolis-Hasting algorithm, the a-priori pdf are as follows: $\mathcal{U}(10,100)$ for $\alpha_r$, $\mathcal{N}(0,0.3)$ for $\beta_r$ and $\mathcal{N}(\mu_{A0/\bar{A}},\sigma_{A0/\bar{A}})$ for $(A_0/\bar{A})_r$. Following the statistics obtained from the HydroSWOT and Pepsi databases $\mu_{A0/\bar{A}} = 0.73$, $\sigma_{A0/\bar{A}} = 0.21$.

Given $A_{0,r}^{(0)}$ and the measurements $(Z_{r,0}, W_{r,0})$, the corresponding bathymetry profile $b_r^{(0)}$ is explicitly obtained, see Section 2.1. These bathymetry values are the "prior" plotted in figures 13 and 12.

The target bathymetry values are those employed in the various calibrated reference flow models (HEC-Ras, LisFlood etc) which have been performed to obtain the synthetic data available in the Pepsi 1 and Pepsi 2 datasets, see Durand et al. (2016); Frasson et al. (Submitted) and references therein. In the figures 11, 12 and 13, the "true" (target) values are represented by the red dots. The latter are computed from the effective rectangular values of the unobserved lowest cross-section $A_0$ $(W = W_0, H_0 = Z_0 - b)$. These "true" values of bathymetry $b$ and $A_0$ are available at Reference Data Scale only (see Section 2).

**4.3.2 First guess $Q_{in,p}^{(0)}$**

Given $(A_{0,r}, (\alpha_r, \beta_r))^{(0)}$, the first guess $Q_{in,p}^{(0)}$ is explicitly obtained from the algebraic flow model (5). First guess values $Q_{in,p}^{(0)}$ are plotted in Fig. 13 ("prior" curve) for rivers within the learning partition and in Fig. 12 ("prior" curve) for rivers outside the learning partition. In both cases, these low Froude estimations $Q_{in,p}^{(0)}$ better catch the variations of the true values than $Q^{(ANN)}$ (indicated as "ANN" on the figures). The estimation $Q_{in,p}^{(0)}$ may be viewed as a physically-consistent correction of the purely data driven estimation $Q^{(ANN)}$. *Remark*. If a mean value of $Q^{(true)}$ is known for a given period (e.g. a week, a month), then one can make fit this information with $Q_{in,p}^{(0)}$. Then, with such an information $Q_{in,p}^{(0)}$ would already be an excellent estimation. However in ungauged rivers, such mean value is unavailable.

**5 Physically-based estimations of $(Q(x,t), A_0(x), K(x; h(x,t)))$ by Variational Data Assimilation (VDA)**

VDA aims at estimating the unknown "input parameters" of the Saint-Venant flow model which are the time-dependent discharge at inflow $Q_{in}(t)$, the bathymetry $b(x)$ (equivalently $A_0(x)$) and the friction coefficient $K$ ($K$ is parametrized as indicated in (4)). The data (WS measurements) are employed as follows. The elevation values $Z$ are used in the cost function which measures the misfit, see (8), $W$ is used to build up the efficient cross-sections geometry of the Saint-Venant flow model, see Section A, while the slope values $S$ are used in the algebraic flow model only, see (5).

**5.1 The VDA formulation**

The employed VDA formulation is the one developed in Larnier et al. (2020a) with a few improvements. At the observation scale, the discrete unknown "parameters" of the dynamic flow model (Saint-Venant's equations) reads:

$$c = (Q_{in,0}, ..., Q_{in,P}; b_1, ..., b_R; (\alpha_1, \beta_1), ..., (\alpha_R, \beta_R))^T \tag{7}$$

[Figure]

**Figure 8.** FlowChart of the complete inversion algorithm. Input data are the WS measurements $(Z, W, S)$ at SwReachSc (SWOT like). Prior is $\mathcal{A}\,(km^2)$ only (from eg. HydroSHEDS database). After the learning period (eg. one year), effective bathymetry $b(x)$ (equivalently $A_0(x)$) and low-Froude effective friction coefficients $K(x; h)$ are estimated (plus discharge values at the fine CompGridSc scale). Next, during the operational period, given newly acquired WS measurements, discharges are computed in real CPU-time at SwReachSc.

[revised manuscript text omitted]

It is worth noticing that this calculation and its consequences remain of course true for the non-inertial version of the Saint-Venant model classically employed in hydrodynamic river flow codes.

A consequence of this "equifinality issue" is the following: at each minimization iteration in the VDA process (see Section 5 for details), the "model constraint" (B1) is satisfied by an infinity of flow states values $(A, Q)$ characterized by the parameter $K$. In other words, the flow model (B1) constrains the inverse problem solution $(Q_{in}(t), A_0(x), K(h))$ up to a multiplicative factor only. This property is of course observed in the numerical results (see Section 6.3): the space-time variations of discharge are accurately infered, however with a bias. This bias depends to the prior information introduced in the inverse method. The introduced priors in the present VDA formulation are detailed in Section 5.3.

In the case the bathymetry $b(x)$ is given (therefore $A_0(x)$), the re-scaled unknown $(A, Q^*)$ does not satisfy the flow model anymore. In other words, if the bathymetry is given, the inverse problem based on the Saint-Venant model (B1) may be well posed. Moreover it has been demonstrated in Garambois and Monnier (2015); Brisset et al. (2018) that a single measurement of bathymetry (i.e. at a single location) enables an accurate estimation of the bathymetry along a relatively long river portion.

In the case, a (simple scalar) mean value of $Q$ is known (e.g. seasonal or annual value), the bias issue is solved ! The inverse problem may be well-posed. The numerical results confirm this assertion, see Section 6.3: in such a case, the estimations of $Q(x, t)$ are accurate, without bias.

[revised manuscript text omitted]

**8 Conclusions**

This study proposes the first combined deep learning - data assimilation approach to infer river discharge values from altimetry measurements only (here synthetic SWOT like data with 1 day repeat). The resulting algorithm, named HiVDI (Hierarchical Variational Discharge Inference), is an important improvement of the former version presented in Larnier et al. (2020a); Frasson et al. (Submitted). The considered data are synthetic flow observations generated from outputs of various calibrated flow models for 29 rivers portions available in the reference Pepsi 1 and 2 datasets, Durand et al. (2016); Frasson et al. (Submitted)). The only ancillary data of the method is a local drainage area extracted from the HydroSHEDS dataset Lehner et al. (2008). The elaborated algorithm relies on: a deep neural network providing a first estimation of $Q$ which is next improved by a low Froude flow model (algebraic flow model). The resulting discharge estimation is good for ungauged rivers at the "SWOT reach scale" (SwReachSc) if presenting discharge values within the learning partition ( $\approx 20 - 40\%$ nRMSE with the present perfect data). For river discharges outside the learning partition, the time variations are very well captured but a bias remains; this bias is inherent to the automatic learning approach. However, one should generally be able to classify rivers by (rough) classes in terms of mean discharge from eg. the GRADES database, Lin et al. (2019). As a consequence, for a majority of rivers portions, one should be able to perform the estimations within the learning partition and not outside. Since the employed input data are non correlated, neither in space nor in time, the accuracy is not affected if considering less frequent data (eg with a few days frequency).

Next, an advanced Variational Data Assimilation (VDA) method enables the estimation of accurate space-time variations of $Q(x,t)$ for any location and any time at fine scale; however the potential bias may remain. It is mathematically shown that this bias cannot be removed if defining the estimations by inverting the classical hydro models only. However any mean value of $Q$ (eg. seasonal or annual) enables to remove the bias, therefore providing accurate estimations. In short, the estimation is based on automatic learning and the inversion of two hierarchical flow models. Given a representative WS measurements set (eg. during a complete year), the method results on two calibrated flow models of hierarchical complexity (algebraic and ) for each river portion. Next, given newly acquired WS measurements, the algebraic flow model is accurate enough ($\approx 15 - 35\%$ nRMSE for perfect data) and low-cost enough to provide in real time the estimation of $Q$ at km scale and at the hours observation scale. The HiVDI algorithm is implemented into the open-source computational software DassFlow Larnier et al. (2020b). In forthcoming studies, this algorithm should be evaluated on the basis of the SWOT simulator data (considering both the Cal-Val orbit and the nominal orbit). Once these more advanced evaluations done, the HiVDI algorithm may be employed to estimate both discharge and bathymetry from the forthcoming SWOT datasets for ungauged or poorly gauged rivers.

[revised manuscript text omitted]

---

## Referee Comment (RC2) · Anonymous Referee #2 · 5 Nov 2020

This study aims at proposing the hybrid Neural Network (NN) – variational data assimilation algorithm to estimate river discharge from simulated SWOT like data. Such methodological studies are very important and of the scope of the NPG. In addition, investigating the potential benefits of satellites prior to the launches is quite useful to improve satellite missions further. However, I think the present manuscript has some fatal issues that should be solved prior to publication. The authors seemed to investigate the method that would not be applicable to the real ungauged river basins as I elaborate below. I am compelled to suggest this manuscript be rejected.

[Major Issues] 1. As described, the SWOT-based estimation of river discharge is useful

for ungagged or poorly gauged river basins (P1L14). However, the authors used "too rich" basin information. They used dA (difference in cross section), W (river width), S (slope), and A (cross section) to estimate Q (discharge) by NN (P8L166). The physical based models, which were also used to mimic observation data, simulates Q based on dA, W, S, and A with only one major uncertainty parameter: frictions of river channel. Namely, there is one equation and one uncertain parameter. Solving this problem is too very easy for NN. Consequently, the present experimental setting of NN was very confusing to me. It is usually impossible to use the cross section A because the cross section under the river surface is unobservable by satellites. The challenge for realistic applications is to estimate Q without using A. 2. The authors assumed unrealistic daily SWOT observation data while real satellite revisits 1-4 times per 21 days (P1L22). Consequently, I strongly suggest the authors re-consider experimental design that is applicable to real problems.

[Other Issues] 1. Experimental design is unclear to me. It is better to add a schematic that shows a flow chart of data used in this algorithm. 2. The paper should add more hydrological papers for reference. For example, I found a data-driven estimation of river width from satellite data (Yamazaki et al. 2014). Comparisons to such existing approach would be beneficial to add values of the manuscript.

https://agupubs.onlinelibrary.wiley.com/doi/full/10.1002/2013wr014664

---

## Author Comment (AC6) · 10 Nov 2020

*This study aims at proposing the hybrid Neural Network (NN) – variational data assimilation algorithm to estimate river discharge from simulated SWOT like data. Such methodological studies are very important and of the scope of the NPG. In addition, investigating the potential benefits of satellites prior to the launches is quite useful to improve satellite missions further. However, I think the present manuscript has some fatal issues that should be solved prior to publication. The authors seemed to investigate the method that would not be applicable to the*

*real ungauged river basins as elaborate below. I am compelled to suggest this manuscript be rejected.*

*Major Issues 1.*
*As described, the SWOT-based estimation of river discharge is useful for ungagged or poorly gauged river basins (P1L14). However, the authors used "too rich" basin information. They used dA (difference in cross section), W (river width), S(slope), and A (cross section) to estimate Q (discharge) by NN (P8L166).*

We are so sorry that you have completely misunderstood the addressed inverse problem and the developed method.
The considered information are the measured quantities by SWOT $(dA, W, S)$ (you are right) plus $\mathcal{A}$ the local drainage area (in $km^2$); obviously (or unfortunately !?) not the river cross-section $A$ (in $m^2$)...
Undoubtedly, there is a serious misunderstanding; moreover there was a typing error P8L166.
However, this crucial point was indicated in the abstract, in the general introduction (P2L58), in Section 2 (P6L135), in the ANN description (Section 3), in figures titles and in the general conclusion. The only input information in addition to the SWOT like measurements is $\mathcal{A}$ the local drainage area.
However this was not recalled P8L166; now it is done. Moreover, P8L166: obviously, the knowledge of $dA$ does *not* imply the knowledge of $A_0$... (typing error which have now been corrected).

The employed values of $\mathcal{A}$ are those available in HydroSHEDS (Hydrological data and maps based on SHuttle Elevation Derivatives at multiple Scales).

Moreover, the Artificial Neural Network (ANN) provides a first rough estimation of Q at reach scale only, see Section 3. The latters being next improved by a (low complexity)

algebraic flow model (Section 4) and finally by the inversion of a complete dynamic flow model inversion (Section 5).

We regret that you did not see the answers to RC1 published sept. 25th; these answers would have brought you some additional clarifications and clues on the addressed challenging scientific problem.

In the revised version published on the journal website Oct. 3rd, see https://npg.copernicus.org/preprints/npg-2020-32/npg-2020-32-AC5-supplement.pdf, we have better highlighted the hypotheses in many locations of the manuscript: in the new abstract, in the general introduction, in the data section, in the conclusion, and in almost each section. Also we have included a flowchart of the complete inversion algorithm with the indication of the unique prior $\mathcal{A}$, the SWOT-like input variables and the output variables, see Fig. 8 P19.

*\* The physical based models, which were also used to mimic observation data, simulates Q based on dA, W, S, and A with only one major uncertainty parameter: frictions of river channel. Namely, there is one equation and one uncertain parameter. Solving this problem is too very easy for NN.*

Again, this is obviously not the addressed inverse problem... As indicated throughout the paper (see above), the addressed inverse problem consists to infer : the discharge value $Q(x,t)$ and effective pairs (friction parameter $K(h(x))$, bathymetry $b(x)$ - or equivalently $A_0(x)$).

You are right, if the problem was to solve a single equation with a single parameter, one line of trivial calculation would have been enough.

We recall in Section 5.3 the "Capabilities and limitations of the inversions based on the

flow models only". This section mathematically shows the inversions capabilities from the standard flow models (this includes the basic Manning-Strickler's law of course). To our best knowledge, this basic but very informative analysis is original. Moreover, it nicely explains the obtained bias when inverting physically-informed models if no prior information (eg an accurate mean value of $Q$) is available, see the cited references or the intercomparisons studies [Durand et al. 2016], [Frasson et al., 2020] (submitted).

* Let us provide some additional point-to-point answers below.

When we refer to the manuscript, we mention either page-lines numbers of the original version (you have received) or the page-line numbers of the version published on the journal website Oct. 3rd (https://npg.copernicus.org/preprints/npg-2020-32/npg-2020-32-AC5-supplement.pdf ).

* *Consequently, the present experimental setting of NN was very confusing to me. It is usually impossible to use the cross section A because the cross section under the river surface is unobservable by satellites. The challenge for realistic applications is to estimate Q without using A.*

It is unfortunately a misunderstanding of the considered inverse problem and the developed methods. Please, refer to the previous answer.

* *2. The authors assumed unrealistic dailySWOT observation data while real satellite revisits 1-4 times per 21 days (P1L22). Consequently, I strongly suggest the authors re-consider experimental design that is applicable to real problems.*

You are right, the considered SWOT-like data are synthetic, 1-day repeat. They cover

however a very large rivers sets with very different flow characteristics. Moreover this responds to an important science issue, at the forefront of the current Discharge Algorithm Working Group (https://swot.jpl.nasa.gov/documents/4050/).

As mentioned in our RC1 point-to-point answers, the first three months after launch, the instrument will be on a 1-day revisit period; this is the important "fast-sampling" Cal-Val phase, see [Rodriguez, JPL, 2012]. This is the context of the present study. This point was not sufficiently highlighted. Now, it is much better indicated throughout the manuscript, including in the new abstract, in the general introduction and conclusion, and of course in the data section too,( see the new version https://npg.copernicus.org/preprints/npg-2020-32/npg-2020-32-AC5-supplement.pdf ).

Note that if considering the nominal SWOT orbit (which will provide data with 21 days revisit period, depending on the latitude), the scientific challenge which consists to solve the ill-posed inverse problem for ungauged rivers posed by the mission remains the same (see Section 5.3 of the manuscript).
In this case, the time validity of the discharge estimation equals the wave travelling time through the river portion (roughly, a few hours to a day, depending on the case), see eg. [Tourian et al. 2017], [Brisset et al 2018], [Larnier et al. 2020] (with the identifiability map concept in particular). This point is well understood now.
The present remark has been added in the dedicated new section 3.4 entitled "On the sensitivity of the estimations with respect to error measurements or data frequency".

Moreover let us point out that in the present ANN, the concept of spatial correlation or time correlation between examples does not exist. Indeed, the ANN input variables are $dA$, $W$, $S$ and $\mathcal{A}$; one "example" corresponds to a set of $(4 + 1)$ values which are point-wise, snapshots. No space or time correlations exist between two "examples". As a consequence, the ANN does not "see" the potential space and time correlations

in the dataset. In our case, if considering less frequent observations (eg. with few days frequency), but of course with similar volume and quality of data, the accuracy of the trained ANN would be similar. We have investigated this assertion for a frequency of $5$ days (results not shown here). As expected the obtained accuracy were of same order of magnitude than those presented in Table 2 (new version of manuscript). Obviously, in this case (eg. with $21$ days revisit) and for the reason previously mentioned (see the identifiability map concept introduced in [Brisset et al. 2018], [Larnier et al. 2020]), the discharge estimations remain valid for a few hours - a day around the observation instant only.

*[Other Issues] 1. Experimental design is unclear to me. It is better to add a schematic that shows a flow chart of data used in this algorithm.*

Thank you for your remark. Following this remark and RC1 comment, we have added a flowchart of the complete inversion algorithm with the indication of the prior, the input variables and the output variables. Please, see Fig. 8 p19.

*2. The paper should add more hydrological papers for reference. For example, I found a data-driven estimation of river width from satellite data (Yamazaki et al. 2014). Comparisons to such existing approach would be beneficial to add values of the manuscript.*

Thank you for mentioning this reference. This reference has already been cited in the new version ( Oct. 3rd, see https://npg.copernicus.org/preprints/npg-2020-32/npg-2020-32-AC5-supplement.pdf, see P 5 L40) to mention a potential river width database (the Global With Database). However, this reference does not address at all the present inverse problem.

Moreover, we have added too: [Paiva et al., WRR 2015], [Tarpanelli et al., IEEE 2018], [Lin et al., WRR 2019].